# Overexpression of human BAG3$^{P209L}$ in mice causes restrictive cardiomyopathy

Kenichi Kimura [1,11], Astrid Ooms[1,11], Kathrin Graf-Riesen[1,11], Maithreyan Kuppusamy [2], Andreas Unger[3], Julia Schuld[4], Jan Daerr[4], Achim Lother[5], Caroline Geisen[1], Lutz Hein [5], Satoru Takahashi [6], Guang Li [7], Wilhelm Röll[8], Wilhelm Bloch[9], Peter F. M. van der Ven [4], Wolfgang A. Linke [3], Sean M. Wu [7], Pitter F. Huesgen [2,10], Jörg Höhfeld[4], Dieter O. Fürst[4], Bernd K. Fleischmann [1✉] & Michael Hesse [1✉]

An amino acid exchange (P209L) in the HSPB8 binding site of the human co-chaperone BAG3 gives rise to severe childhood cardiomyopathy. To phenocopy the disease in mice and gain insight into its mechanisms, we generated humanized transgenic mouse models. Expression of human BAG3$^{P209L}$-eGFP in mice caused Z-disc disintegration and formation of protein aggregates. This was accompanied by massive fibrosis resulting in early-onset restrictive cardiomyopathy with increased mortality as observed in patients. RNA-Seq and proteomics revealed changes in the protein quality control system and increased autophagy in hearts from hBAG3$^{P209L}$-eGFP mice. The mutation renders hBAG3$^{P209L}$ less soluble in vivo and induces protein aggregation, but does not abrogate hBAG3 binding properties. In conclusion, we report a mouse model mimicking the human disease. Our data suggest that the disease mechanism is due to accumulation of hBAG3$^{P209L}$ and mouse Bag3, causing sequestering of components of the protein quality control system and autophagy machinery leading to sarcomere disruption.

[1] Institute of Physiology I, Life and Brain Center, Medical Faculty, University of Bonn, Bonn, Germany. [2] Central Institute for Engineering, Electronics and Analytics, ZEA-3, Forschungszentrum Jülich, Jülich, Germany. [3] Institute of Physiology II, University of Münster, Münster, Germany. [4] Department of Molecular Cell Biology, Institute for Cell Biology, University of Bonn, Bonn, Germany. [5] Institute of Experimental and Clinical Pharmacology and Toxicology, Division II, Faculty of Medicine, University of Freiburg, Freiburg, Germany. [6] Department of Anatomy and Embryology, Faculty of Medicine, University of Tsukuba, Tsukuba, Japan. [7] Stanford Cardiovascular Institute, Stanford University School of Medicine, Stanford, CA, USA. [8] Department of Cardiac Surgery, University of Bonn, Bonn, Germany. [9] German Sport University Cologne, Department of Molecular and Cellular Sport Medicine, Cologne, Germany. [10] CECAD, Medical Faculty and University Hospital, and Institute of Biochemistry, Faculty of Mathematics and Natural Sciences, University of Cologne, Cologne, Germany. [11] These authors contributed equally: Kenichi Kimura, Astrid Ooms, Kathrin Graf-Riesen. ✉email: bernd.fleischmann@uni-bonn.de; mhesse1@uni-bonn.de

Bcl2-associated athanogene 3 (BAG3) is a co-chaperone and proteostasis factor, which mediates the degradation of unfolded proteins under stress, such as mechanical strain. In humans, a single base mutation in *BAG3* leads to an amino acid exchange (P209L) causing severe childhood restrictive cardiomyopathy, muscular dystrophy, respiratory insufficiency, and peripheral polyneuropathy[1] resulting in increased mortality by the second decade (OMIM MFM6 #612954).

The BAG3 protein is strongly expressed in striated muscle and localizes to thin filament-anchoring Z-discs. A critical target protein of BAG3 is the actin-crosslinking protein filamin C (FLNC)[2,3]. BAG3 mediates the degradation of mechanically unfolded and damaged forms of FLNC through chaperone-assisted selective autophagy (CASA). Thus, the co-chaperone exerts key proteostasis functions under mechanical strain. This is supported by the analysis of BAG3$^{-/-}$ mice, which exhibit disrupted Z-discs in striated muscle cells resulting in growth retardation and fulminant myopathy after birth in response to muscle usage[4]. Interestingly, in humans with the BAG3 (P209L) mutation, Z-disc disruption and formation of protein aggregates have been reported[1] pointing to a possible disease mechanism in agreement with recent in vitro work[5,6]. However, this aspect needs to be explored in vivo but this has not been possible because of the lack of a mouse model displaying the typical features of the human disease. For instance, insertion of the homologous point mutation into the mouse *Bag3* gene did not produce any obvious phenotype[7]. Likewise, cardiomyocyte (CM)-specific overexpression of human BAG3 harboring the P209L point mutation (hBAG3$^{P209L}$) in mice caused mild dilated cardiomyopathy with late-onset but no aggregate formation, sarcomere disintegration, or the prototypic restrictive cardiomyopathy[8]. The reasons underlying the lack of a typical phenotype is unclear, but the degree of BAG3$^{P209L}$ transgene expression was not determined in this study. Thus, the need for an animal model that mimics the human pathology is critical because the pathophysiology of this severe, life-threatening cardiac disease cannot be studied in humans at the cellular and molecular level due to the general lack of heart biopsies from patients.

Herein we report the establishment of several transgenic mouse lines expressing hBAG3$^{P209L}$ in striated muscle cells. We find that these humanized mouse models mimic key pathophysiological features of patients enabling us to explore the underlying molecular disease mechanisms and its rescue via an AAV-mediated gene therapy approach.

## Results

### Generation and analysis of αMHC-BAG3$^{P209L}$ mice.
To gain insights into the pathomechanisms underlying BAG3$^{P209L}$-induced cardiomyopathy, we generated transgenic mouse lines with CM-specific overexpression of either hBAG3$^{WT}$-eGFP (αMHC-BAG3) or hBAG3$^{P209L}$-eGFP (αMHC-BAG3$^{P209L}$) (Supplementary Fig. 1a). This strategy was chosen to investigate and to rule out potential adverse effects of hBAG3$^{WT}$-expression. Mice of both transgenic lines were viable, had normal litter sizes, and did not display any obvious phenotype. Macroscopically, the hearts did not display any differences at 3- and 10-weeks of age and cardiac dry weights to tibia lengths did not differ between αMHC-BAG3$^{WT}$, αMHC-BAG3$^{P209L}$, and control (WT) mice (Supplementary Fig. 1b). Both transgenic mouse lines displayed eGFP expression in their hearts with postnatal onset, but the expression pattern in αMHC-BAG3$^{P209L}$-transgenic hearts was more heterogeneous and patchy (Fig. 1a). Quantification of eGFP expression in sections from αMHC-BAG3$^{WT}$ and αMHC-BAG3$^{P209L}$ hearts revealed that 83.13 ± 2.58% of CMs ($n = 3$) expressed BAG3$^{WT}$, while expression of BAG3$^{P209L}$ was found in only 36.28 ± 5.56% of CMs ($n = 3$). The observed difference in transgene penetrance between the two mouse lines is most likely due to positional effects of the random integration of the transgene into the genome (18 copies of the transgene for αMHC-BAG3$^{WT}$ and 32 for αMHC-BAG3$^{P209L}$) and/or silencing of the promoter by epigenetic effects. Therefore, we isolated and sorted eGFP$^+$ CMs from αMHC-BAG3$^{WT}$ and αMHC-BAG3$^{P209L}$ mice and determined by immunoblotting the degree of overexpression of hBAG3$^{P209L}$ in comparison to endogenous mouse Bag3. This ratio was found to be 1.27 ± 0.66 ($n = 3$) for αMHC-BAG3$^{WT}$, and 1.04 ± 0.26 ($n = 3$) for αMHC-BAG3$^{P209L}$ (Supplementary Fig. 1c, d).

### Aggregate formation and disruption of sarcomers in αMHC-BAG3$^{P209L}$ CMs.
Microscopic analysis of isolated single CMs from adult αMHC-BAG3$^{P209L}$ mice demonstrated that BAG3$^{P209L}$-eGFP expression led to almost complete Z-disc disintegration, accompanied by aggregation of α-actinin and BAG3$^{P209L}$ (Fig. 1b). Quantification of isolated CMs showing either formation of BAG3-aggregates or aggregates and disturbances of α-actinin cross striation at 10 weeks of age revealed that the majority (64.66 ± 1.57%) of eGFP$^+$ CMs in BAG3$^{P209L}$-eGFP expressing hearts were affected. In contrast, only 10.67 ± 1.15% and 3.33 ± 0.58% of CMs from αMHC-BAG3$^{WT}$ and WT hearts, respectively, displayed some structural abnormalities (Supplementary Fig. 1e).

Next, we extended this analysis to BAG3-interacting proteins and cytoskeletal components. The distribution of both the BAG3 client FLNC and the BAG3 partner protein SYNPO2 was severely altered in BAG3$^{P209L}$-eGFP$^+$ CMs, as both proteins displayed loss of their typical cross-striated pattern and formation of small aggregates (Fig. 1c). Also, HSPA8 and HSPB8, chaperone partners of BAG3, formed aggregates upon BAG3$^{P209L}$ expression (Fig. 1d). In contrast, BAG3$^{P209L}$ expression in CMs did not affect the sarcomeric protein titin, which is not under control of BAG3-mediated protein quality control (Supplementary Fig. 1f). Indeed, regions with large BAG3$^{P209L}$ aggregates were devoid of titin (Supplementary Fig. 1f, arrow), whereas the intermediate filament protein desmin displayed structural alterations (Supplementary Fig. 1f) due to hBAG3$^{P209L}$ expression. These findings illustrate (i) that there is a profound dominant-negative impact of BAG3$^{P209L}$ on sarcomeric architecture and (ii) that different Z-disc components may depend on different protein quality control (PQC) systems or they are less affected by the aggregation cascade initiated by BAG3$^{P209L}$.

Ultrastructural analysis revealed severe sarcomeric disruption upon BAG3$^{P209L}$ expression in adult CMs, with sarcomeric lysis and readily detectable formation of electron-dense aggregates (Fig. 1e). Also, mitochondria in CMs were disordered to some extent compared to WT controls (Fig. 1e). These striking structural disarrangements were exclusively found in hBAG3$^{P209L}$-eGFP$^+$ CMs, but not in hBAG3P209L-eGFP$^-$ neighboring CMs of the same hearts, as demonstrated by immunogold staining for eGFP (Fig. 1f). Staining against eGFP labeled structures and aggregates identical to those labeled by the BAG3 antibody that recognizes both hBAG3 and endogenous mouse Bag3 (Fig. 1g, Supplementary Fig. 2a). This suggested that the aggregates were composed of both transgenic BAG3$^{P209L}$ and endogenous Bag3. Aggregate formation was often detected adjacent to either intact (I), or partly (P) or severely (S) disintegrated sarcomeres (Fig. 1g), thereby strongly resembling the phenotype in human patients[9]. We also analyzed the composition of the aggregates by immunogold EM and found that they contained BAG3, but only negligible amounts of α-actinin (Fig. 1g), fully in line with our immunofluorescence analysis (Fig. 1b). Additional staining against titin, FLNC, desmin, and β-actin revealed that in BAG3$^{P209L}$ CMs, titin was exclusively found in sarcomeres, whereas FLNC and desmin were also detected in

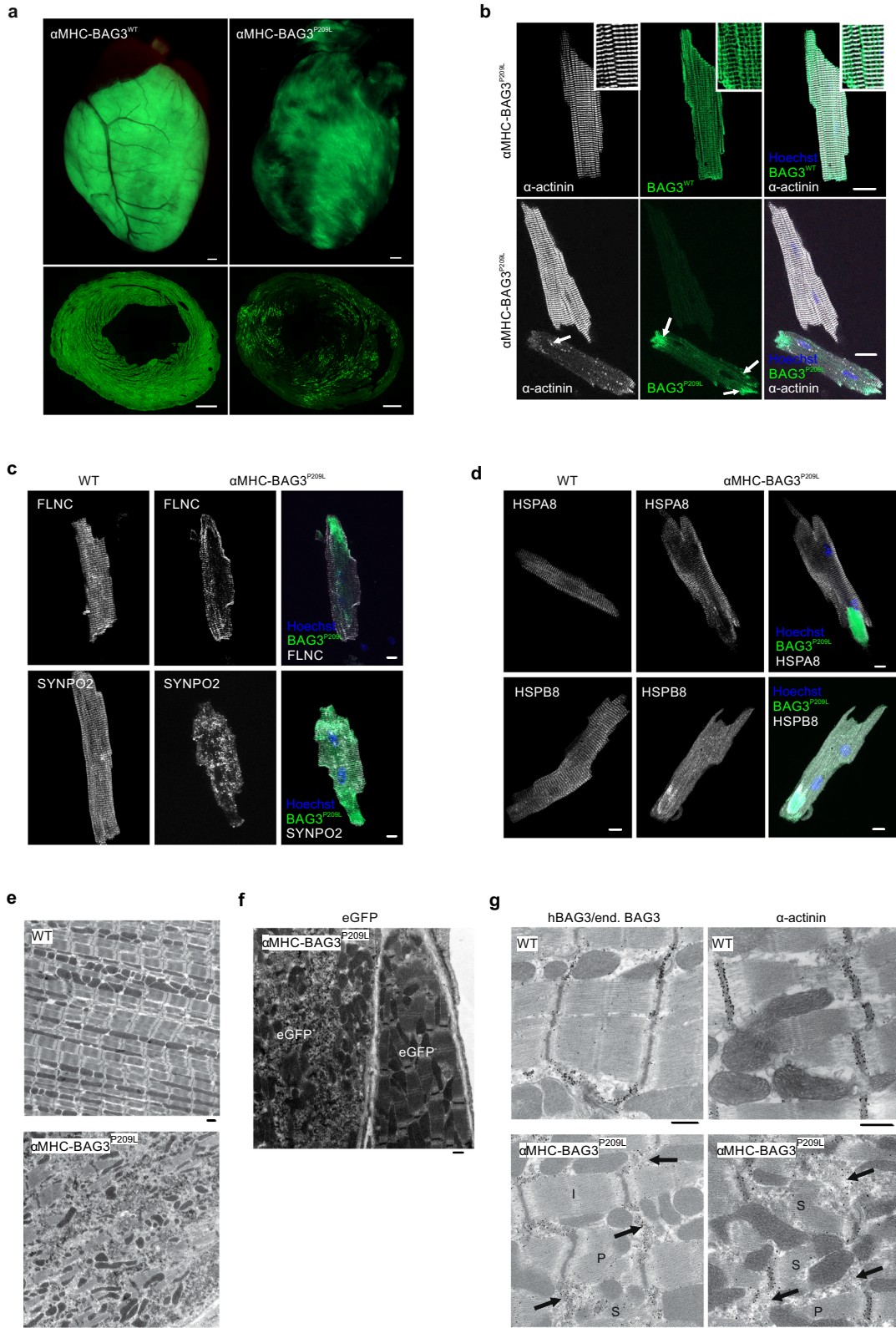

electron-dense aggregates (Supplementary Fig. 2b). In contrast, β-actin localized solely to Z-discs and intercalated discs, but not to aggregates (Supplementary Fig. 2c). Although hBAG3$^{P209L}$-eGFP$^+$ CMs displayed pathological features resembling those observed in patients, the corresponding mice neither presented signs of cardiomyopathy nor strongly compromised heart function and early mortality. This was most likely due to the low degree of penetrance (36.28 ± 5.56% of CMs) of the transgene.

**Overexpression of human BAG3$^{P209L}$ causes restrictive cardiomyopathy.** Because of the obvious limitations of our αMHC-

**Fig. 1 Characterization of αMHC-BAG3^WT and αMHC-BAG3^P209L mice. a** Representative hearts and cross-sections from transgenic αMHC-BAG3^WT and αMHC-BAG3^P209L mice. BAG3^WT-eGFP was expressed in the majority of CMs, while BAG3^P209L-eGFP showed a patchy expression pattern. Scale bars: 500 μm. **b** Langendorff-isolated CMs from 10-week-old αMHC-BAG3^WT, and αMHC-BAG3^P209L mice stained for α-actinin (white) displaying Z-disc localization of BAG3^WT (inserts: zoomed regions), but structural disintegration in BAG3^P209L expressing CMs (**b**, arrows). While BAG3^WT (green, top panel) formed a grid-like structure (zoomed region, top panel), BAG3^P209L (green, bottom panel) was found to form also aggregates of different size (arrows, bottom panel). Scale bar: 20 μm. **c, d** Langendorff-isolated CMs from 10-week-old WT, and αMHC-BAG3^P209L mice were stained for FLNC, SYNPO2 (**c**), HSPA8, and HSPB8 (**d**). BAG3^P209L (green) formed large aggregates in CMs from αMHC-BAG3^P209L mice. Scale bars: 10 μm. **e** Ultrastructural analysis of hearts from 16-week-old WT and αMHC-BAG3^P209L mice; note loss of sarcomeric structures and disarrangement of mitochondria in αMHC-BAG3^P209L CMs. Scale bars: 1000 nm. **f** Immunogold stainings (black dots) of an ultrathin section αMHC-BAG3^P209L mice for eGFP. **g** Immunogold staining for eGFP, hBAG3/endogenous Bag3, and α-actinin with formation of aggregates in CMs from BAG3^P209L mice (arrows) accompanied by different stages of sarcomere disintegration. **a–g** All the experiments were repeated three times from three independent biological replicates with similar results. I, intact sarcomere, P, partly disintegrated sarcomere, S, severely disintegrated sarcomere. WT, wild-type, Scale bars: 1000 nm.

BAG3^P209L mouse model described above, we subsequently generated a transgenic mouse line enabling the conditional expression of hBAG3^P209L-eGFP under the control of the CAG-promoter (Supplementary Fig. 3a). This promoter was chosen because of its known strong expression in muscle tissue[10]. The construct was inserted into the Rosa26 locus of mouse G4-embryonic stem cells (ESCs) by using zinc-finger nucleases and a transgenic mouse line named Tg(CAG-flox-hBAG3^P209L-eGFP) was derived from a positive ESC-clone. This line was bred to homozygosity with PGK-Cre transgenic mice (Supplementary Fig. 3a), because these displayed a more consistent phenotype, whereas heterozygous mice had a quite variable phenotype.

Our initial characterization of this humanized mouse model revealed that only 50% of the expected number of homozygous PGK-Cre/CAG-flox-hBAG3^P209L-eGFP mice (referred to as CAG-BAG3^P209L) from heterozygous matings were born, which points to an increased embryonic lethality. After birth, we found that CAG-BAG3^P209L mice were substantially smaller than their littermates (Fig. 2a), displaying growth retardation. The body-weight was significantly reduced by 2 weeks of age and amounted to 12.6 ± 4.1 g by 5 weeks of age ($n = 6$; controls: 23.4 ± 1.8 g, $n = 14$, Fig. 2b). Because of the reduced body weight, the animals had to be euthanized at the age of 5 weeks; 12% of mice died between 2 and 5 weeks of age (Supplementary Fig. 3b). We found no difference in the heart-weight to tibia length ratio, whereas the heart weight to bodyweight ratio was increased in CAG-BAG3^P209L mice (Supplementary Fig. 3c). Reminiscent of the human pathology and starting at around 2 weeks of age, the mice displayed prominent signs of skeletal muscle weakness with some variations in onset and severity of the overall phenotype.

To better understand the striking phenotype of the mice, we performed macroscopic and microscopic analysis of eGFP-fluorescence of 1- to 5-week-old CAG-BAG3^P209L mouse hearts. The transgene was found to be expressed in 75.3 ± 14.1% of ventricular CMs at 2 weeks of age, 76.7 ± 14.5% at 3 weeks, and 74.9 ± 8.9% at 5 weeks of age ($n = 3–4$ mice; Fig. 2c, Supplementary Fig. 3d, e). Transgene expression in hearts was detected exclusively in CMs, as demonstrated by immunofluorescence staining (Supplementary Fig. 3f). Moreover, transgene expression proved to be muscle-specific, as BAG3^P209L-eGFP expression was predominantly visible in striated muscle (Supplementary Fig. 4a).

To determine if the ratio of transgenic hBAG3 to endogenous mouse Bag3 protein in hearts from 5-week-old CAG-BAG3^P209L mice was comparable to the estimated 1:1 ratio of BAG3^P209L to BAG3 in patients with the P209L mutation, we performed immunoblotting, yielding a ratio of 1.05 ± 0.08 (Supplementary Fig. 4b). Next, we performed histological analysis of 1-, 3-, and 5-week-old hearts from CAG-BAG3^P209L and control mice. Left and right ventricles displayed similar dimensions (Fig. 2d, Supplementary Fig. 4c, d), but we found that the number of enlarged CMs (Fig. 2e, arrows) increased with age. In 5-week-old

mice large, swollen nuclei and the formation of vacuoles in CMs became evident (Fig. 2e, arrowheads, and insets). Quantification yielded a significant increase in CM size starting at 1 week and progressing further at 3–5 weeks of age (Fig. 2f). This was accompanied by disruptions of the regular structure of cardiac muscle fibers seen in controls (Fig. 2e), possibly indicating fibrotic alterations. To test this, we determined the extent of fibrosis in CAG-BAG3^P209L mice by Sirius-red staining. Indeed, enhanced cardiac fibrosis was already detected at 2 weeks of age and it further increased significantly in 3- and 5-week-old CAG-BAG3^P209L mice, when compared to controls (Fig. 3b). At 5 weeks of age, fibrosis amounted to more than 40% (Fig. 3a, b). As cardiac fibrosis can be caused by collagen deposits replacing apoptotic cells, we stained cardiac sections from 2- and 5-week-old CAG-BAG3^P209L and control mice for the apoptosis marker activated caspase 3 (Supplementary Fig. 4e, f) and counted the number of positive CMs. There was a gradual increase of CM apoptosis from 2 to 5 weeks of age (Supplementary Fig. 4f), which is concordant with the observed increase in fibrosis (Fig. 3b). At 2 and 5 weeks of age, the number of apoptotic CMs was significantly higher in CAG-BAG3^P209L mice compared to controls (Supplementary Fig. 4f), suggesting partial replacement fibrosis.

Immunofluorescence staining against BAG3 revealed that transgene-expressing CMs from 3- to 5-week-old CAG-BAG3^P209L mice displayed BAG3^P209L-eGFP containing aggregates (Fig. 3c–e, Supplementary Fig. 5a–g) and that these also contained endogenous mouse Bag3 (Supplementary Fig. 5a). This was accompanied by structural changes of the microfilament system and the intermediate filaments in CMs (Fig. 3c, d), which progressed with age (Supplementary Fig. 5b, c). Similar to our findings in CMs from αMHC-BAG3^P209L transgenic mice, immunostainings revealed that the CASA client FLNC and the BAG3 partner protein SYNPO2 were found in aggregates in CAG-BAG3^P209L mice (Fig. 3e, f), while proteins not associated with CASA, such as titin, were not detected in the aggregates (Supplementary Fig. 5d). Remarkably, the intermediate filament (IF) protein vimentin, which was not expressed in control CMs, progressively accumulated in aggregates in CMs from CAG-BAG3^P209L mice with increasing age (Fig. 3g, Supplementary Fig. 5e), possibly indicating a switch to a fetal gene program. We also found prominent infiltration of CD45+ immune cells into the myocardium (Fig. 3h, Supplementary Fig. 5f); at least in part, these were CD68+ macrophages phagocytosing the apoptotic CMs (Supplementary Fig. 5g).

Next, we performed ultrastructural analysis of hearts from 4-week-old CAG-BAG3^P209L mice and found severe sarcomere disruption and lysis in CMs with formation of electron-dense aggregates (Fig. 4a). Mitochondria in CMs were disordered compared to controls, possibly due to the severe Z-disc alterations (Fig. 4a). Immunogold staining for eGFP revealed BAG3^P209L-

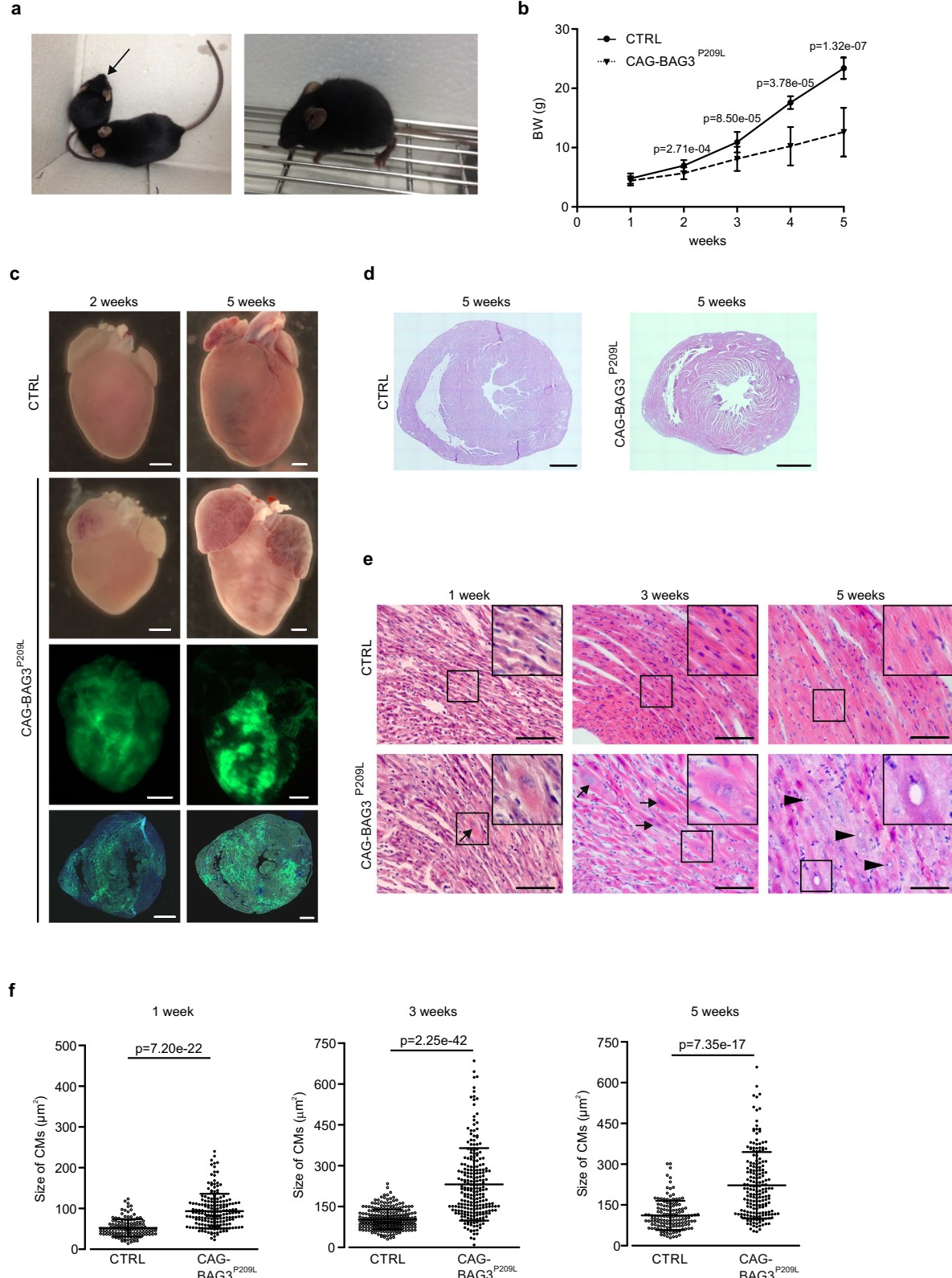

eGFP localization to electron-dense aggregates between partially or completely disintegrated sarcomeres (Fig. 4b). Thus, the ultrastructural data corroborate our immunohistochemical findings and reveal that overexpression of BAG3[P209L] in CMs leads to aggregate formation and disintegration of sarcomeres.

Given the prominent alterations of CM morphology and the strong cardiac fibrosis, we next investigated heart function using echocardiography in 21–25-day-old CAG-BAG3[P209L] mice, because at this age, the mice could still endure this procedure. We found typical features of restrictive cardiomyopathy[11]: The left ventricular inner diameter was smaller in CAG-BAG3[P209L] mice compared to WT littermates yielding an increase in relative wall thickness, reduced stroke volume, and cardiac output despite preserved ejection fraction (Fig. 4c–e, Supplementary Fig. 6a,

**Fig. 2 CAG-BAG3^P209L mice display growth retardation and cardiac alterations compared to littermate controls. a** Typical growth difference between a CAG-BAG3^P209L and a CTRL mouse (left, arrow). The 5-week-old homozygous mouse displays signs of kyphosis and muscular dystrophy (right). **b** Growth curve of CTRL and CAG-BAG3^P209L mice from 1- to 5-weeks. Mean ± SD. N (1 week) = 23 CTRL and 17 CAG-BAG3^P209L mice; n (2 weeks) = 23 CTRL and 15 CAG-BAG3^P209L mice; n (3 weeks) = 16 CTRL and 21 CAG-BAG3^P209L mice; n (4 weeks) = 7 CTRL and 10 CAG-BAG3^P209L mice; n (5 weeks) = 14 CTRL and 6 CAG-BAG3^P209L mice. Two-sided Student's T-test for each age. **c** Representative hearts from 2- and 5-week-old CAG-BAG3^P209L and CTRL mice. Hearts of CAG-BAG3^P209L mice show nondilated ventricles with marked dilation of both atria. Scale bars: 1mm. The experiments were repeated three times from three independent biological replicates with similar results. **d** Histological analysis of hearts from 5-week-old CAG-BAG3^P209L and CTRL mice by HE-staining. Scale bars: 1mm. The experiments were repeated three times from three independent biological replicates with similar results. **e** HE-staining of hearts from 5-week-old homozygous CAG-BAG3^P209L and CTRL mice. CMs start to enlarge at 1 week (arrows), number of enlarged CMs increased with age. At 5 weeks, enlarged CMs had swollen nuclei (arrowheads); boxed areas are shown as insets. Scale bar: 100 μm. **f** Quantification of CM square area in cross-sections from hearts of control and CAG-BAG3^P209L mice at 1-, 3-, and 5-weeks. Mean ± SD. N (1 week) = 135 CTRL CMs, n = 190 CAG-BAG3^P209L CMs; n (3 weeks) = 262 CTRL and 234 CAG-BAG3^P209L CMs; n (5 weeks) = 120 CTRL and CAG-BAG3^P209L CMs. Two-sided Student's T-test. CTRL, control mice (siblings of CAG-BAG3^P209L mice which are either WT, PGK-Cre, or CAG-flox-hBAG3^P209L).

---

Supplementary Table 1). Reduced relaxation of the left ventricle during diastole caused impaired diastolic filling (LVPWd/LVIDd, Fig. 4c, d, Supplementary Fig. 6a, Supplementary videos 1–2). Doppler measurements revealed a restrictive mitral inflow (E/A ratio > 2.5) in 4 / 8 CAG-BAG3^P209L- transgenic versus 0 / 6 control mice (Fig. 4f, g). Pulmonary artery pressure, as derived from pulmonary artery measurements[12] was elevated in CAG-BAG3^P209L mice, and accordingly also right ventricular function was impaired (Fig. 4h, i, Supplementary Table 1). At 5 weeks of age, CAG-BAG3^P209L mice either died or suffered from heart failure (cardiac output severely reduced, Supplementary Fig. 6a) and had to be euthanized, as explained above.

**Changes in the PQC system and increased autophagy in CAG-BAG3^P209L mice.** To gain deeper molecular insights into the pathophysiological mechanisms occurring in hearts from CAG-BAG3^P209L mice, we performed single-cell and tissue transcriptomic analysis, as well as high-resolution proteomic analysis. Since the expression level of hBAG3^P209L-eGFP is heterogeneous across the myocardium and the observed pathologic phenotypes (and presumably the changes in gene expression) are only evident in eGFP+ CMs, we performed single-cell RNA-Seq analysis of eGFP+ CMs in control and CAG-BAG3^P209L mice at the earliest stage of disease onset (2 weeks of age). These experiments showed that the ratio of hBAG3 to mouse Bag3 was in the range of 0.5- to 2-fold, in accordance with both the measured eGFP fluorescence intensity (dim and bright) of individual CMs (Supplementary Fig. 7a) and the protein levels determined by immunoblotting (Supplementary Fig. 4b). To expand our single eGFP+ CM-expression data, we also performed differential expression analysis for our scRNA-seq data from clusters of 33 control (n = 2 mice) and 80 bright (n = 2 mice) individual CMs to determine changes in transcription by KEGG pathway analysis. We found 35 significantly downregulated genes (Supplementary Data 1); some of these pointed to an impairment of energy metabolism pathways, since TCA cycle, oxidative phosphorylation pathways, as well as genes involved in fatty acid elongation, were affected (Supplementary Fig. 7b, Supplementary Data 2). Other pathways, such as the fetal gene program, indicative of heart failure, were not activated at this stage of disease onset. To further assess the global transcriptional derangement in failing CAG-BAG3^P209L hearts, we performed tissue RNA-Seq analysis from hearts of 5-week-old CAG-BAG3^P209L and control mice, since single-cell RNA-Seq was not feasible due to the prominent cardiac fibrosis. This analysis revealed that a total of 3804 genes were differentially expressed, of which 1950 were upregulated and 1854 down-regulated in homozygous CAG-BAG3^P209L mouse hearts compared to controls (Fig. 5a, Supplementary Data 3). Among the most strongly upregulated genes were components of the protein quality control system (e.g. Hspa1a, Hspa8, Cryab), fibrotic

pathways (e.g., CTGF, Galectin-3, Endothelin-1) and of the fetal CM gene program (e.g., Nppa, Myh7). Following GO-analysis for KEGG pathway terms with either the set of upregulated or downregulated genes, the most affected genes were revealed in energy metabolism pathways such as the TCA cycle and oxidative phosphorylation (Fig. 5b). Also, genes involved in cardiac muscle contraction were downregulated, as would be expected considering the loss of sarcomeres. Upregulated genes were related to the KEGG terms lysosome, phagosome, and endocytosis, indicating accumulation of many proteins involved in these protein quality control pathways (Fig. 5b, Supplementary Data 4).

Given that BAG3 is involved in the post-translational control of protein homeostasis, we next analyzed the heart proteome of CAG-BAG3^P209L and control mice at the age of 2 and 5 weeks using an unbiased label-free shotgun proteomics approach. This identified 2351 proteins in the hearts of 2-week-old mice, but only a single protein, Myeloid leukemia factor 1, showed significant accumulation according to our stringent criteria for determining significant changes in abundance (two-sample t-test, Benjamini–Hochberg adjusted FDR < 0.05) in CAG-BAG3^P209L mice compared to controls (Supplementary Data 5). In contrast, 5-week-old mice revealed massive alterations of the heart proteome: Out of 1347 identified proteins, 351 proteins were either significantly up (217 proteins) or downregulated (134 proteins) (Fig. 5c, Supplementary Data 6). GO enrichment analysis of the proteins with significantly lower abundance in 5-week-old CAG-BAG3^P209L mice showed that most belonged to major metabolic pathways of the mitochondria, including oxidative phosphorylation and TCA cycle (Supplementary Fig. 8a). In addition, also proteins required for cardiac muscle contraction, including (e.g., TNNT2 and TPM1) were depleted. In contrast, protein subunits of the proteasome, focal adhesion, and tight junctions, as well as heat shock proteins (CRYAB, HSPB7) and proteins involved in autophagy/phagosome formation, were accumulating in the hearts of 5-week-old CAG-BAG3^P209L mice (Fig. 5d). Consistent with the physiological phenotype and transcript abundance data, also proteins involved in fibrotic pathways (Galectin-3) and associated with cardiomyopathies, the fetal cardiac gene program (LMNA, MYH7, NPPA) and vasopressin-regulated water reabsorption, as a sign of heart failure, were accumulating. Overall, the changes in protein abundance in 5-week-old mice in comparison to transcript abundance showed a significant correlation with the RNA data (Pearson correlation coefficient 0.68, −lg10(pval) > 15.65), suggesting that most protein changes were the result of changes in transcription (Supplementary Fig. 8b). Importantly, we noticed that proteins listed as BAG3-interactors in Biogrid[13] displayed on average a significantly higher accumulation than all other proteins (Mann–Whitney-U test p-val > 0.001), consistent with a BAG3-driven protein accumulation process (Fig. 5e).

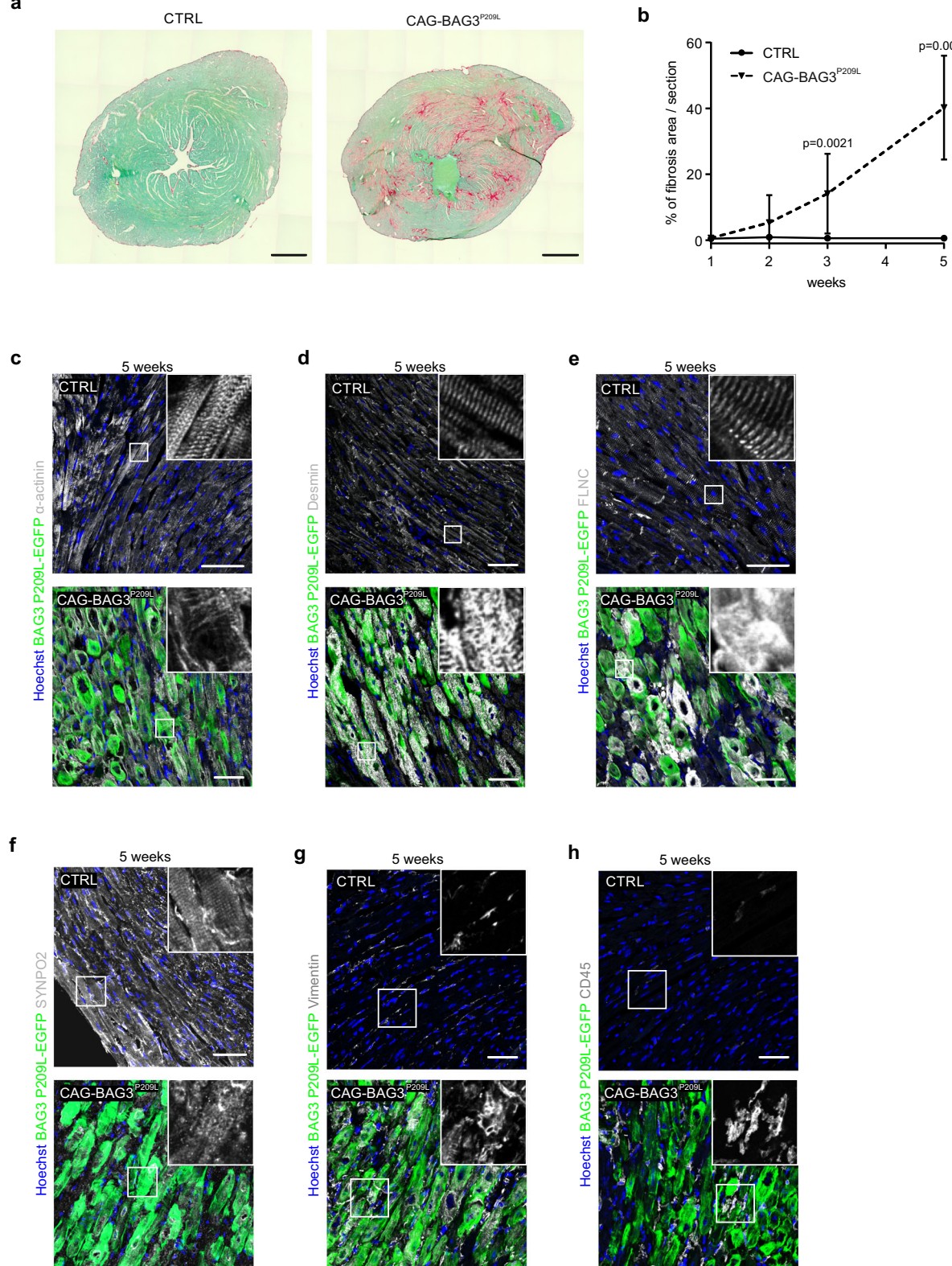

**Fig. 3 Cardiac fibrosis and formation of aggregates in CMs from CAG-BAG3$^{P209L}$ mice. a** Analysis of cardiac fibrosis by Sirius red and fast green stainings: red collagen staining indicates massive fibrosis in hearts from CAG-BAG3$^{P209L}$ mice. **b** Quantification of the fibrotic area in CAG-BAG3$^{P209L}$ mice and CTRL. Mean ± SD. $N$ (1 week) = 3 CTRL and 3 CAG-BAG3$^{P209L}$ mice; $n$ (2 weeks) = 3 CTRL and 6 CAG-BAG3$^{P209L}$ mice; $n$ (3 weeks) = 10 CTRL and 13 CAG-BAG3$^{P209L}$ mice; $n$ (5 weeks) = 4 CTRL and 6 CAG-BAG3$^{P209L}$ mice. One-way ANOVA. **c–h** Sections of hearts from 5-week-old CAG-BAG3$^{P209L}$ and CTRL mice were stained for α-actinin (**c**), desmin (**d**), FLNC (**e**), SYNPO2 (**f**), vimentin (**g**), and CD45 (**h**). In BAG3$^{P209L}$–eGFP (green) CMs the formation of large aggregates was observed. Scale bars: 50 µm. **c–h** The experiments were repeated three times from three independent biological replicates with similar results. CTRL, control mice (siblings of CAG-BAG3$^{P209L}$ mice, which are either WT, PGK-Cre, or CAG-flox-hBAG3$^{P209L}$).

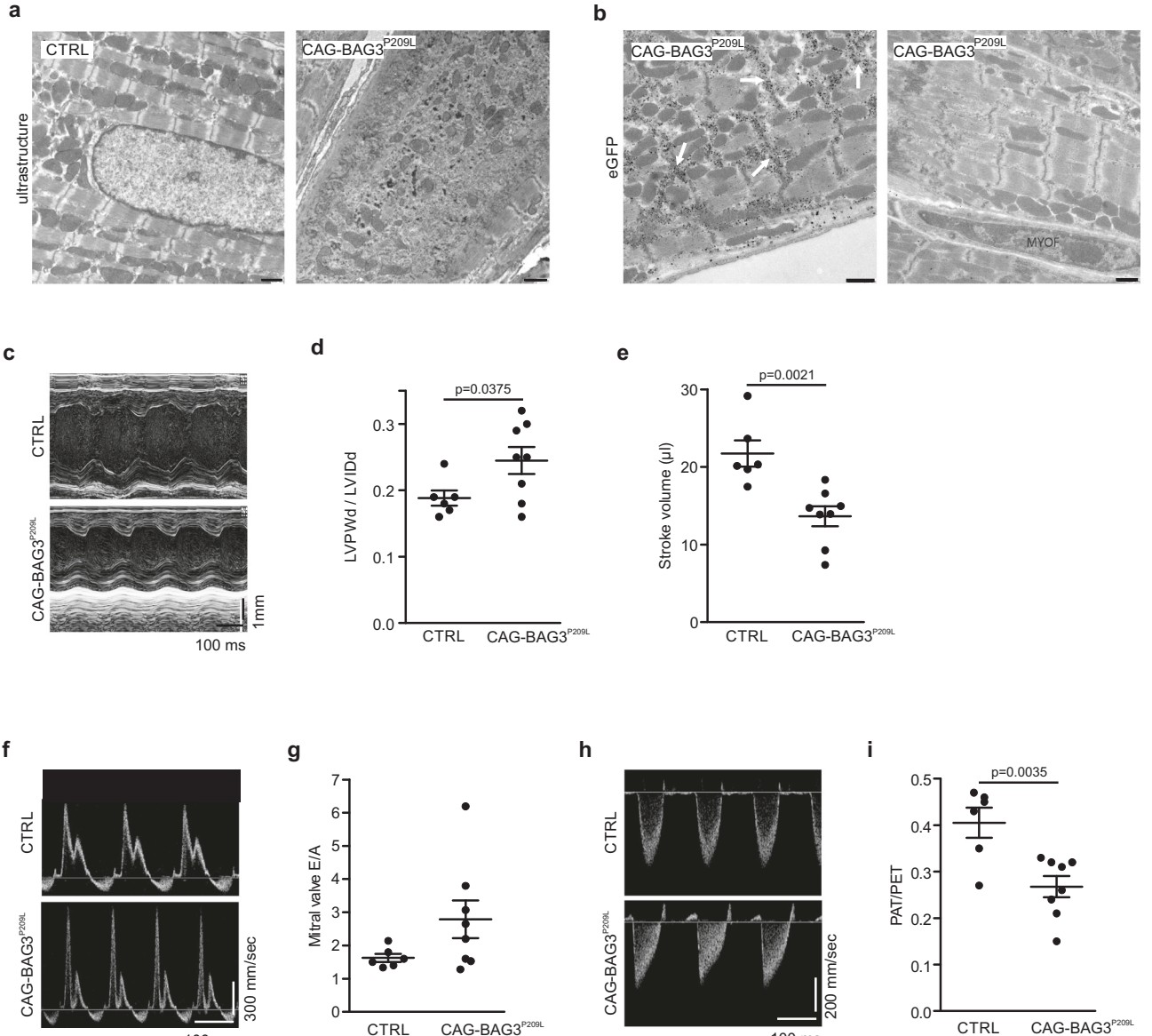

**Fig. 4 Disintegration of sarcomeres and development of restrictive cardiomyopathy in CAG-BAG3[P209L] mice. a, b** Ultrastructural analysis of hearts from 4-week-old control and CAG-BAG3[P209L] mice (**a**) and (**b**) immunogold staining for eGFP, revealing disintegration of sarcomeres and formation of aggregates in CMs from CAG-BAG3[P209L] mice (arrows). MYOF = myofibroblast. Scale bars: 1000 nm. The experiments were repeated three times from three independent biological replicates with similar results. **c–i** Echocardiography of 4-week-old CAG-BAG3[P209L] and CTRL mice: **c–i** Increased relative wall thickness (**c, d**) and reduced stroke volume (**c, e**) were found in CAG-BAG3[P209L] mice; data were obtained from parasternal M-Mode of the left ventricle. Tendency of increased mitral valve E/A ratio (**f, g**) and increased pulmonary artery acceleration/ejection time (PAT/PET; **h, i**) were found in CAG-BAG3[P209L] mice; data were derived from Doppler flow measurements. LVPWd left ventricular posterior wall during diastole, LVIDd left ventricular inner diameter during diastole. Mean ± SEM. (**d, e, g, i**) $n = 6$ CTRL and 8 CAG-BAG3[P209L] mice. Two-sided Student's *T*-test. CTRL, control mice (siblings of CAG-BAG3[P209L] mice, which are either WT, PGK-Cre, or CAG-flox-hBAG3[P209L]).

As our proteome analysis suggested accumulation of small heat shock proteins and increased autophagic processes, we next validated these observations by immunoblot analysis of total heart protein from 5-week-old CAG-BAG3[P209L] and control mice. In accordance with the increased number of autophagic vesicles detected in the EM analysis (Figs. 1e, 4a, b), we found strong induction of SQSTM1 and LC3B in CAG-BAG3[P209L] mice (Fig. 5f). This suggested a strong impact on autophagic flux, leading to the formation of more autophagosomes, as has also been described in patient samples[14]. In addition, endogenous mouse Bag3, as well as BAG3-associated proteins, such as CRYAB, were found to accumulate compared to controls (Fig. 5f),

underscoring our immunofluorescence results. Likewise, we confirmed accumulation of the small heat shock proteins HSPB6 and HSPB7 (Fig. 5f).

Thus, RNA-Seq and proteomics data demonstrated accumulation of components of the protein quality control system and the autophagy machinery in hearts from CAG-BAG3[P209L] mice.

**BAG3[P209L] has altered protein solubility, and decreased mobility.** We next investigated the biochemical properties of the BAG3[P209L] protein to better understand the disease mechanisms. First, we analyzed whether its interaction with partner proteins

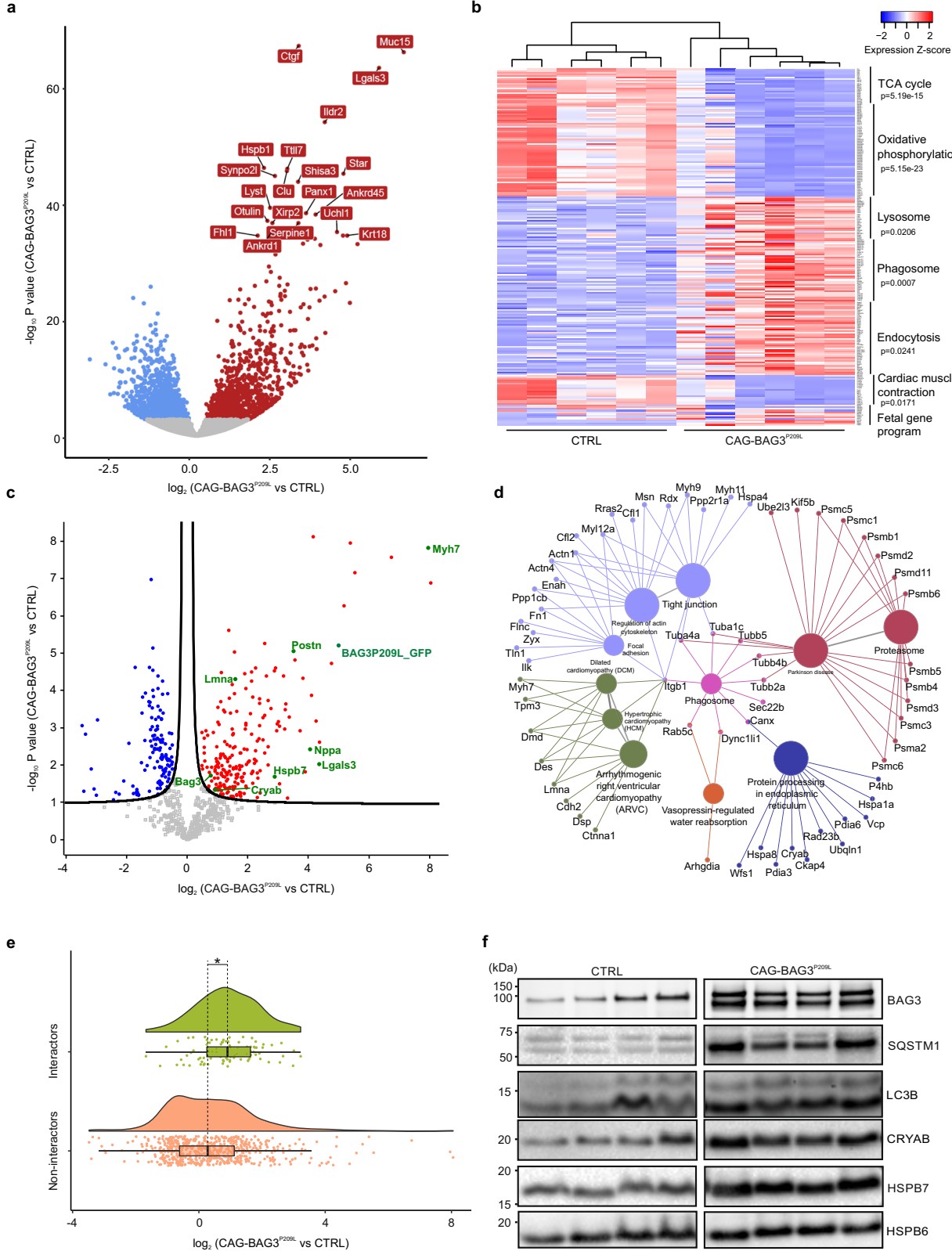

was impaired by performing in vitro binding studies with purified components. We focused on the binding of BAG3 to HSPB8, which has been reported to be mediated by two conserved IPV (Ile-Pro-Val) motifs[15], of which the second one is affected by the P209L mutation (Supplementary Fig. 9a). Surprisingly, BAG3[P209L] was found to associate with HSPA8, FLNC, and HSPB8 similar to BAG3[WT] (Supplementary Fig. 9b–d). Since

binding was unaltered, conformational changes might occur in BAG3, potentially explaining its tendency to aggregate. Therefore, purified hBAG3[WT] and hBAG3[P209L] were subjected to limited proteolysis by trypsin treatment (Supplementary Fig. 9e). We found that the mutation caused several changes in the fragment pattern following treatment (Supplementary Fig. 9e, open arrowheads), pointing to an altered tertiary/quaternary structure

**Fig. 5 RNA-Seq-based transcriptome and proteome analysis of hearts from CAG-BAG3[P209L] and control mice. a, b** RNA-Seq analysis revealed differential gene expression in hearts from 5-week-old CAG-BAG3[P209L] mice ($n = 6$ mice) compared to CTRL ($n = 6$ mice). Volcano plot showing false discovery corrected $q$-values and $\log_2$(fold)-changes. Blue and red dots depict genes depleted and enriched respectively in CAG-BAG3[P209L] mice compared to CTRL. Names of the 20 most differentially expressed genes are shown. **b** Same hearts as in (**a**). Gene expression heatmap for significantly ($P < 10^{-6}$ Bonferroni step down corrected for multiple comparison) enriched KEGG pathways identified by ClueGO (two-sided hypergeometric test) and selected genes from a fetal cardiac gene program found in pathological hypertrophy (*Myh6, Atp2a2, Atp2b4, Hand2, Gata6, Mef2c, Nkx2-5, Gata4, Acta1, Lmna, Myh7, Nppa, Nppb*). **c** Volcano plot from proteome analysis of hearts from 5-week-old CAG-BAG3[P209L] vs CTRL mice. Blue and red dots depict the proteins significantly depleted and accumulated respectively in the CAG-BAG3[P209L] mice compared to the CTRL (Two-sided Student's $T$-test $p$-value <0.05, FDR < 0.05, S0 = 0.1). **d** KEGG term analysis of proteins with significantly increased abundance in CAG-BAG3[P209L] mice compared to CTRL mice. **e** Raincloud plots illustrate the change in protein abundance in hearts of CAG-BAG3[P209L] compared to CTRL mice for proteins known to interact with hBAG3 ($n = 86$) and all others ($n = 629$). In the box plot, center lines show the medians with box plot limits indicate the 25th and 75th percentiles. Box plot whiskers extend to 1.5x interquartile range from the 25th and 75th percentiles. Asterisk indicates a significant difference between both populations (Two-sided Mann–Whitney rank sum test, $p$-value 5.7E−6). **f** Immunoblot analysis of hearts from 5-week-old mice revealed the presence of hBAG3 and a significant increase in the amount of mouse Bag3 in CAG-BAG3[P209L] mice in comparison to CTRL mice. CAG-BAG3[P209L] mice had nearly as much hBAG3 as mouse Bag3. Note significant increases in the expression levels of the heat shock proteins HSPB7 and α-B crystallin, and the autophagy marker p62. Expression was normalized to the expression level of α-actin. The data were obtained from four independent biological replicates and not technically repeated. CTRL, control mice (siblings of CAG-BAG3[P209L] mice, which are either WT, PGK-Cre, or CAG-flox-hBAG3[P209L]).

of hBAG3[P209L]. We also investigated the solubility and heat sensitivity of purified hBAG3[WT] and hBAG3[P209L] (Supplementary Fig. 9f). These experiments revealed that the amount of insoluble hBAG3[P209L] significantly increased following heat treatment, pointing to a more aggregation-prone conformation of the mutant cochaperone. Thus, the P209L mutation does not per se alter the binding properties of the protein, but rather its susceptibility to aggregate formation.

To corroborate the increased tendency of BAG3[P209L] to form aggregates in vivo, we performed differential centrifugation of cardiac protein extracts (Fig. 6a) from adult CAG-BAG3[P209L] and homozygous PGK-Cre/CAG-flox-hBAG3[WT]-eGFP mice (referred to as CAG-BAG3[WT], Supplementary Fig. 10a–h). Strikingly, hBAG3[P209L]-eGFP was mainly found in the insoluble pellet fraction, underscoring the strong predisposition of hBAG3[P209L] to form aggregates in vivo. Besides, in the presence of hBAG3[P209L] endogenous mouse Bag3 was driven into the insoluble fraction, indicating that the mutant protein also sequesters mouse Bag3 into aggregates (Fig. 6b). Similar results were obtained for CMs isolated from αMHC-BAG3[WT] or αMHC-BAG3[P209L] transgenic mice (Supplementary Fig. 10i).

As hBAG3[P209L] was preferentially detected in the insoluble fraction, we next tested its mobility by performing fluorescence recovery after photobleaching (FRAP) experiments with hBAG3[P209L] and hBAG3[WT] control CMs. After bleaching of BAG3[WT]-eGFP (Fig. 6c), a fast recovery rate with a half-life of $21 \pm 5$ s and a mobile fraction of $73 \pm 8\%$ was measured, illustrating that the protein undergoes highly dynamic interactions at the Z-disc (Fig. 6d, e). In strong contrast, exchange rates for hBAG3[P209L] were significantly reduced, in terms of both the fluorescence half-life, which could not even be calculated within the time of the experiment, and the mobile fraction ($13 \pm 3\%$; Fig. 6d, e).

**AAV-mediated knockdown of hBAG3 mitigates the disease phenotype.** Given the prominent cardiac phenotype of our mouse model and its resemblance to the human disease, we wondered, whether we could halt or even reverse its key features using gene therapy. CAG-BAG3[P209L] mice were treated at P15 with a single intrajugular injection of AAV/rh10, harboring a cassette for the expression of either a shRNA targeting *hBAG3[P209L]* or a randomly scrambled shRNA (Supplementary Fig. 11a). AAV/rh10 was chosen because of its strong tropism and expression in striated muscle[16]. We analyzed the mice at P37 and found that the body weight in the hBAG3-shRNA group was significantly higher compared to the mice treated with scrambled shRNA (Fig. 7a), indicating the effectiveness of our therapy (Fig. 2b). Next, the transduction rate of

AAV/rh10 in CMs was determined, and mCherry expression was detected in 69.8 ± 10.3% of CMs after treatment with AAV hBAG3-siRNA and in 63.5 ± 11.6% after treatment with AAV scrambled siRNA (Supplementary Fig. 11b), whereas non-CMs were found to be mCherry-negative. EGFP-fluorescence was analyzed as a readout for the pathological protein expression, and we detected in most animals of the hBAG3-shRNA group a striking decrease of hBAG3[P209L]-eGFP fluorescence, whereas mCherry fluorescence intensity was comparable in both treatment and control groups (Supplementary Fig. 11b, c). This was further corroborated by immunoblot analysis of hearts from hBAG3-shRNA and scrambled shRNA-treated mice, which revealed a prominent reduction of hBAG3[P209L] protein (Fig. 7b). Moreover, the fluorescence intensities of mCherry and hBAG3[P209L]-eGFP were found to be inversely correlated (Supplementary Fig. 11b, c). In parallel with the reduction of the hBAG3[P209L]-eGFP fluorescence, we also noticed a significant reduction of the number of aggregates in CMs of the hBAG3-shRNA treated group. Furthermore, even regular cross-striation could be detected in the vast majority of CMs after staining for α-actinin, desmin, or FLNC (Fig. 7c–e), and CM swelling and immune cell infiltration were strongly reduced (Fig. 7f) in the hBAG3-shRNA treated group. As a further proof of the rescue effect, the prominent cardiac fibrosis seen in CAG-BAG3[P209L] mice was significantly decreased upon knockdown of hBAG3[P209L] (Fig. 7g, h).

Thus, AAV-shRNA-based reduced hBAG3[P209L]-eGFP expression results in mitigation of the cardiac phenotype.

## Discussion

Herein, we demonstrate that our CAG-BAG3[P209L] mouse model phenocopies the devastating human cardiac disease. In fact, we detected a progressive early-onset restrictive cardiomyopathy, as reported for the patients suffering from BAG3[P209L] myofibrillar myopathy[1,9,17]. The mice developed increasingly severe symptoms of heart failure accompanied by growth retardation starting shortly after birth and, therefore, requiring their euthanization at 5 weeks of age. In patients with the BAG3[P209L]-mutation, the restrictive cardiomyopathy is due to extensive fibrosis causing a stiffening of the myocardium[18] and an impairment of diastolic filling of the ventricles. Congruent with the human pathology we detected in CAG-BAG3[P209L] mice cardiac fibrosis starting at 3 weeks and becoming very extensive by 5 weeks of age. It manifested as a diffuse interstitial and replacement fibrosis triggered by the loss of CMs by apoptosis. Mouse models for restrictive cardiomyopathy are rare and mainly based on mutations in proteins determining the passive properties of CMs, such

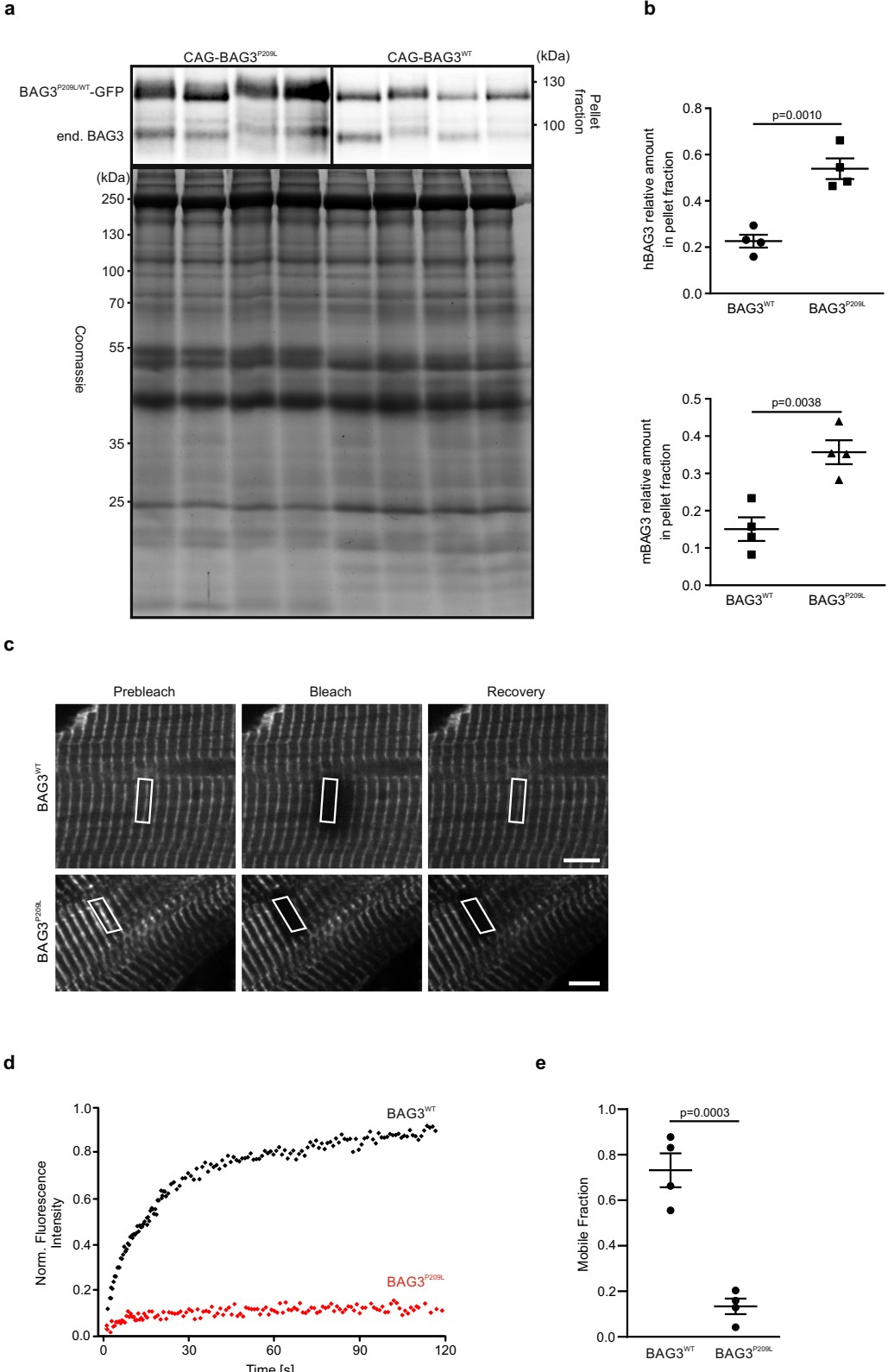

as troponin I or troponin T. This is different in our mouse model, where dysfunction of an important component of the protein quality control system causes CM apoptosis, activation of a fetal gene program and pronounced replacement fibrosis. Interestingly, another mouse line with CM-specific overexpression of hBAG3[P209L] was reported[8] to lack an overt cardiac phenotype or typical changes of CMs except for an accumulation of pre-

amyloid oligomers at the adult stage. One reason potentially underlying this difference could be the ratio of hBAG3[P209L] to mouse Bag3, which in our mouse models appears comparable to that in human patients, who are heterozygous for the dominant-negative P209L mutation. Similarly, a mouse model harboring the homologous P209L point mutation in mouse Bag3 protein (P215L) did not present an obvious cardiac phenotype, even

**Fig. 6 Reduced solubility and dynamics of hBAG3P209L. a** Protein extracts from hearts of 5-week-old CAG-BAG3P209L and CAG-BAG3WT mice were prepared and analyzed by differential centrifugation and subsequent immunoblotting. BAG3P209L-eGFP was predominantly found in the insoluble pellet fraction and seemed to have sequestered endogenous mouse Bag3 into the insoluble fraction. **b** Quantification of the amount of hBAG3 and endogenous mBag3 in the pellet fraction. Mean ± SEM. n = 4 hearts per group. Two-sided Student's T-test. **c–e** Langendorff-isolated BAG3-eGFP-expressing CMs from hearts from 10-week-old αMHC-BAG3WT and αMHC-BAG3P209L mice were analyzed by FRAP. Shown is a time series of fluorescence micrographs indicating the localization of hBAG3WT and hBAG3P209L at Z-discs. **c** Regions of interest (indicated by white boxes) were followed over time and are shown before bleaching ("Prebleach"), immediately after bleaching ("Bleach"), and after 1 min of fluorescence recovery ("Recovery"). Scale bars: 5 µm. **d** Typical recovery profiles of hBAG3WT and hBAG3P209L. Note very fast recovery of hBAG3WT, whereas hBAG3P209L fluorescence is essentially not recovering at all during the time of observation. **e** Bar plot summarizing mobile fractions of hBAG3WT (73%) and hBAG3P209L (13%) in Z-discs. Note the considerably higher dynamics of hBAG3WT in comparison to hBAG3P209L. Mean ± SEM. n = 8 CMs per group. Two-sided Student's T-test.

when bred to homozygosity[7], indicating subtle differences between human and mouse BAG3 proteins.

We next explored the disease mechanisms and focused first on the CMs, which are also severely affected in the human pathology. In both of our mouse models, CAG-BAG3P209L-GFP and aMHC-BAG3P209L-GFP, we observed the second hallmark of the human disease, namely aggregation of proteins in cytoplasmic inclusions that correlated directly with the BAG3P209L protein expression and sarcomere disintegration. The aggregates consisted mainly of BAG3, αB-crystallin, desmin, and vimentin, whereas FLNC and α-actinin formed separate aggregate entities with a morphology distinct from those containing hBAG3P209L (Fig. 3c, e). Our data imply that hBAG3P209L expression directly causes formation of aggregates leading to a collapse of the BAG3-associated proteostasis network over time, resulting in sarcomere dissolution as a consequence of muscle usage. At 2 weeks of age, a strong hBAG3P209L-eGFP expression was evident in a minority of CMs, which were swollen and in which aggregates were already formed. At this stage, the disease phenotype was still relatively mild, which was supported by the RNA-Seq and proteomics data. However, for the next three weeks, more CMs displayed strong BAG3P209L-eGFP expression, a massive formation of aggregates, and cell death resulting in the striking cardiac phenotype. We also noticed an impairment of metabolic pathways of the mitochondria, such as oxidative phosphorylation and TCA cycle. These changes could be partly due to a loss of CMs and future experiments need to explore, whether they are primary caused by hBAG3P209L-expression or secondary due to CM loss.

We also explored the impact of the altered biochemical properties of the hBAG3P209L protein and its potential effects on the endogenous Bag3 protein. Our data argue against altered binding properties of hBAG3P209L as the molecular mechanism responsible for the observed phenotype, because hBAG3P209L retained its ability to bind to HSPB8, HSPA8, and FLNC (Supplementary Fig. 8). The proteomics data underscored this, as we could find an enrichment of BAG3-interactors among the upregulated proteins in hearts from CAG-BAG3P209L mice. Thus, the mutant protein retains its ability to associate with partner proteins, including other proteostasis factors and chaperone clients. This is also supported by recent findings demonstrating that binding of hBAG3P209L to Hsp70 and HSPB8 is mostly unaffected and that both IPV domains need to be mutated to abolish HSPB8 binding[5]. This might also explain why BAG3P209L was able to rescue certain phenotypes associated with knockdown of BAG3 in zebrafish[19]. However, when investigating the aggregation properties of hBAG3P209L, we found them strongly increased in BAG3P209L CMs in vivo. This is most likely a key driver of the disease process and validates earlier in vitro findings[5,6] and in vivo findings in zebrafish[19,20]. It is further underscored by the FRAP experiments, which demonstrated reduced mobility of BAG3P209L, explaining the impairment of autophagic flux and hence its contribution to the accumulation of proteins. Thus, transgenic hBAG3P209L-eGFP expression in our mice leads to co-aggregation and sequestration of endogenous Bag3 in CMs. This

causes attenuation of its proteostasis function in muscle resulting in disintegration of myofibril architecture and a collapse of muscle cell protein homeostasis. The ensuing dysfunction of autophagy is corroborated by the increase and accumulation of p62 and LC3B. In addition, the formation of aggregates and lack of their clearing is a reliable sign of impaired protein quality control. Similar results were reported for a zebrafish model in which the bag3 gene was deleted and human BAG3P209L-eGFP was overexpressed[20]. Also, small heatshock proteins, such as αB-crystallin, HSPB6, and HSPB7 did accumulate, as has been described in a heart from a patient[14], and can therefore no longer fulfill their function. Taken together, our findings highlight the essential and multifaceted role of BAG3 in organizing the proteostasis machinery in striated muscle and to protect from mechanical stress. As a consequence, patients suffering from MFM6 should avoid physical exertion such as strenuous exercise. This could lead to progressive muscle damage, as has been observed in other MFM models[21].

To gain additional insight into pathophysiological processes and to explore its therapeutic potential, we have taken advantage of a gene therapy approach by using AAVs to achieve a specific knockdown of the hBAG3P209L-expression. These proof-of-principle experiments illustrated a strong rescue effect, as the phenotypic hallmarks of the disease, such as growth retardation and clinical signs of heart failure were strongly mitigated. At the cellular level, we found that in CMs only a few protein aggregates were detected and that their structural integrity was largely preserved, highlighting again the pathophysiological impact of protein aggregate formation for the disease process. Our AAV-based gene therapy approach demonstrated that it is possible to halt or even reverse the progression of this severe MFM by reducing expression levels of the pathological protein.

Application of this strategy to patients suffering from MFM6 would require the identification of an shRNA specific for the mRNA carrying the point mutation. This is critical, as loss of BAG3 leads to postnatal mortality, at least in mice[4], and has toxic effects in both zebrafish[19,22] and isolated human CMs[23]. The approach has proven challenging[24] in the past and alternatively a micro-RNA based strategy could be used[25]. A more sophisticated but also more complicated approach would involve the AAV-mediated delivery of CRISPR-based gene-editing tools, such as prime-editing tools for correction of the point mutation[26]. Such a strategy should be developed and tested in CMs derived from patient-specific iPS-cells before application in the clinic.

## Methods

### Generation of αMHC-BAG3WT-eGFP and αMHC-BAG3P209L-eGFP constructs.
Human BAG3WT and BAG3P209L cDNAs[3] were cloned in fusion with the eGFP cDNA under the control of the CAG promoter. By restriction of CAG-hBAG3WT-eGFP with EcoRI, blunting, and subsequent digestion with NotI a 2468 bp fragment containing the hBAG3WT-eGFP cDNA was isolated and inserted into the AgeI (blunted) and NotI sites of an αMHC-plasmid. The exchange of a fragment in the hBAG3WT-eGFP cDNA with the equivalent region of the hBAG3P209L cDNA containing the mutation resulted in αMHC-hBAG3P209L-eGFP.

All vectors were verified by sequencing.

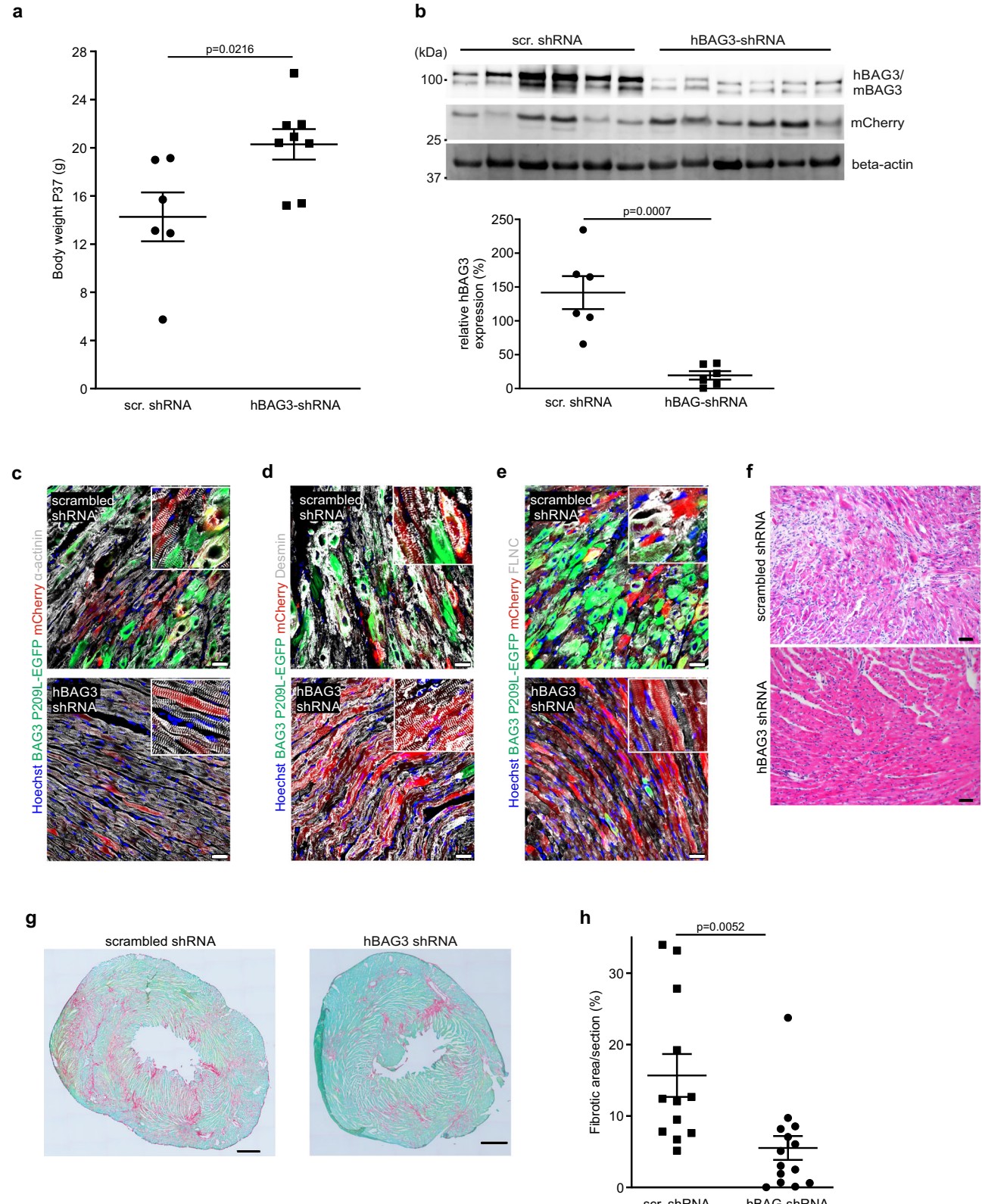

### Generation of Tg(CAG-flox-hBAG3^P209L^-eGFP) and Tg(CAG-flox-hBAG3^WT^-eGFP) constructs.

**Generation of Tg(CAG-flox-hBAG3$^{P209L}$-eGFP) and Tg(CAG-flox-hBAG3$^{WT}$-eGFP) constructs**. Human BAG3$^{WT}$-eGFP and BAG3$^{P209L}$-eGFP driven by the ubiquitous CAG promoter were cloned into the pDonor Rosa26 vector from Addgene (Plasmid #37200). Initially, a hygromycin-stop-cassette flanked by loxP-sequences was inserted downstream of the CAG promoter in the CAG-hBAG3$^{WT}$-eGFP and CAG-hBAG3$^{P209L}$-eGFP plasmid, respectively, by using the Quick&Easy Conditional Knockout Kit-loxP from Gene Bridges. The resulting plasmids were digested with *SnaBI* and *MunI* and 5763 bp fragments containing the 3′ part of the

CAG promoter, the loxP flanked stop cassette and the BAG$^{WT}$-eGFP or BAG$^{P209L}$ expression cassettes were exchanged against the 2233 bp *SnaBI-MunI* fragment of a CAG-eGFP-Neo pDonor Rosa26 vector.

All vectors were verified by sequencing.

**Generation and cultivation of transgenic mESC clones**. The cultivation of G4 hybrid ESCs[27] was performed in Knockout-Dulbecco's modified Eagle's medium (DMEM high glucose, supplemented with 15% v/v fetal calf serum (FCS), 0.1 mM

**Fig. 7 AAV-based gene therapy approach with shRNA against hBAG3[P209L] rescues key features of the disease in vivo. a** Body weight was found to be increased in CAG-BAG3[P209L] mice 3 weeks after a single intrajugular injection of AAV/rh10 hBAG3 shRNA at P15, when compared to CAG-BAG3[P209L] mice treated with AAV/rh10 scr. shRNA. Mean ± SEM.; *n* = 6 CTRL and 8 CAG-BAG3[P209L] mice. Two-sided Student's *T*-test, scr. shRNA = scrambled shRNA. **b** Immunoblot analysis and quantification of protein expression revealed a decrease in the amount of hBAG3 (BAG3 upper band) in CAG-BAG3[P209L] mice treated with AAV/rh10 hBAG3 shRNA in comparison to mice treated with AAV/rh10 scr. shRNA. This correlated inversely with the amount of mCherry expression; for normalization of expression levels β-actin was used. Mean ± SEM. *n* = 6 hearts per group. Two-sided Student's *T*-test. scr. RNA = scrambled RNA. **c–e** Sections of hearts from 5-week-old CAG-BAG3[P209L] mice which were treated with either AAV/rh10 BAG3 shRNA or AAV/rh10 scr. shRNA were stained for α-actinin (**c**), desmin (**d**), FLNC (**e**). Scale bars: 25 μm. **f** HE-staining of hearts from 5-week-old CAG-BAG3[P209L] mice which were treated with either AAV/rh10 hBAG3 shRNA or AAV/rh10 scr. shRNA Scale bar: 100 μm. **c–f** The experiments were repeated three times from three independent biological replicates with similar results. **g** Analysis of cardiac fibrosis by Sirius red and Fast green staining after treatment with either AAV/rh10 BAG3 shRNA or AAV/rh10 scr. shRNA. Shown are sections with 33.20% (scr. shRNA treated) and 9.74% (hBAG3 shRNA treated) fibrotic area. **h** Quantification of the fibrotic area in CAG-BAG3[P209L] mice 3 weeks after treatment with either scr. or hBAG3 shRNA. Mean ± SEM. *n* = 12 CTRL and 14 CAG-BAG3[P209L] mice. Two-sided Student's *T*-test. scr. shRNA = scrambled shRNA.

nonessential amino acids, 2 mg/ml l-Glutamine, 50 μg/ml penicillin/streptomycin, 500 U/ml Leucemia inhibitory factor (LIF), 0.1 mM β-mercaptoethanol. The cells were kept on irradiated neomycin-resistant mouse embryonic fibroblasts (Millipore). For the generation of transgenic mESCs $5 \times 10^6$ cells were mixed with 30 μg of linearized plasmid DNA in PBS, electroporated (250 V and 500 μF and 1 pulse (Bio-Rad, Gene Pulser)) and plated onto two 100-mm dishes. Selection for transgene mESCs started 2 days after electroporation by adding 165 μg/ml G418 to the medium. Resistant colonies were isolated and cultivated onto neomycin-resistant mouse embryonic fibroblasts before they were propagated and analyzed for BAG3[WT]/BAG3[P209L]-eGFP expression.

**Generation of transgenic mice.** Animal experiments were conducted according to the guidelines from Directive 2010/63/EU of the European Parliament on the protection of animals used for scientific purposes. For the generation of transgenic mouse lines, transgenic mESCs with the respective integrated targeting construct and a karyotype of 40 chromosomes were aggregated with diploid morula stage CD-1 embryos[28]. For genotyping of the αMHC-BAG3[WT]-eGFP and αMHC-BAG3[P209L]-eGFP transgenic mice genomic DNA of tail tips was isolated, and PCR was performed using the primers BAG3*for* and BAG3*rev* (see Supplementary Table 2 for primer information). The number of integrations was determined by qPCR of genomic DNA with a probe for the TfrC gene for normalization. There were 18 copies of the transgene in αMHC-BAG3[WT]-eGFP mice and 32 copies in αMHC-BAG3[P209L]-eGFP mice. For genotyping of the Tg(CAG-flox-hBAG3[WT]-eGFP) and Tg(CAG-flox-hBAG3[P209L]-eGFP) transgenic mice genomic DNA of tail tips was isolated and PCR was performed using the primers CAG2-fw and BAG3-h-spez.tev (see Supplementary Table 2 for primer information). The number of integrations was determined by qPCR as one copy of the transgene. For the Tg(CAG-flox-hBAG3[P209L]-eGFP) and Tg(CAG-flox-hBAG3[WT]-eGFP) mouse lines, homozygous mice were defined as having two flox-BAG3[P209L]-eGFP alleles with either one or two deletions of the STOP-cassette, which was achieved by crossing them to PGK-Cre mice. For genotyping of the PGK-Cre mice genomic DNA of tail tips was isolated, and PCR was performed using the primers Pgk1_for and Int-Cre_rev (see Supplementary Table 2 for primer information). Also, in homozygous Tg(CAG-flox-hBAG3[P209L]-eGFP) mice with only one deletion expression was significantly higher than in heterozygous mice with deletion.

Wild-type (WT) mice were defined as non-transgenic littermates of αMHC-BAG3[WT]-eGFP or αMHC-BAG3[P209L]-eGFP mice. Control mice (CTRL) were defined as siblings of homozygous PGK-Cre/CAG-flox-hBAG3[P209L]-eGFP or PGK-Cre/CAG-flox-hBAG3[WT]-eGFP mice with either WT, PGK-Cre, CAG-flox-hBAG3[P209L]-eGFP or CAG-flox-hBAG3[WT]-eGFP genotype. Mice from the homozygous (PGK-Cre/CAG-flox-hBAG3[WT]-eGFP) strain did not display any obvious phenotype despite the strong hBAG3[WT]-eGFP expression in their hearts (Supplementary Fig. 10a–h). Transgenic αMHC-BAG3[WT]-eGFP, αMHC-BAG3[P209L]-eGFP, Tg(CAG-flox-hBAG3[WT]-eGFP), Tg(CAG-flox-hBAG3[P209L]-eGFP), PGK-Cre/CAG-flox-hBAG3[WT]-eGFP, and PGK-Cre/CAG-flox-hBAG3[P209L]-eGFP mice were on a mixed 129S6/SvEvTac × C57BL/6Ncr × CD-1 genetic background.

**Harvesting and fixation of mouse hearts.** Adult mice were heparinized and euthanized by cervical dislocation. The hearts were harvested, and the aorta was cannulated. Hearts were perfused with PBS and afterward with 4% PFA. Further fixation in 4% PFA was done overnight at 4 °C. Isolated embryonic and postnatal hearts were fixed in 4% PFA overnight at 4 °C. Hearts were further incubated in 20% sucrose solution and mounted in O.C.T. compound (Sakura Finetek Europe B.V.).

**Isolation of single CMs by Langendorff dissociation.** CMs were isolated from dissected adult hearts using the Langendorff dissociation[29]. Contraction was prevented by adding 25 mM BDM in all used solutions. The enzyme solution was prewarmed to 37 °C by connecting the perfusion system to a water bath. Hearts were cannulated and connected to the perfusion system. After washing with PBS

and Ca²⁺-free Tyrode's solution hearts were perfused with enzyme solution with a velocity of 2.2 ml/min[30]. After becoming pale and soft hearts were removed from the perfusion system, atria were disconnected, and ventricles were minced mechanically using forceps. The enzyme reaction was stopped with 5 ml perfusion buffer with 5% FBS, 50 μM CaCl₂ and the suspension was filtered through a 100 μm cell filter. Cells were centrifuged at $200 \times g$ for 5 min and the pellet resuspended in 5 ml perfusion buffer with 5% FBS, 50 μM CaCl₂.

**Echocardiography.** Cardiac morphology and function were determined by echocardiography from parasternal and apical views during isoflurane anesthesia (1.5 vol% in oxygen) using a VisualSonics Vevo2100 ultrasound system and Vevo LAB version 3.2.5 software for analysis. Stroke volume was calculated from 2D images (stroke volume = LVIDd³ × ejection fraction)[12]. Pulmonary artery pressure was derived from non-invasive measurement of pulmonary artery flow as described before[31]. The investigator was blinded to genotype.

**Flow cytometry.** Isolated CMs of adult Langendorff-dissociated hearts from αMHC-BAG3[P209L] mice were sorted by flow cytometry at 488 nm using an inFlux v7 Sorter cytometer (Becton-Dickinson). Quantification of eGFP⁺ CMs was done by analysis of *n* = 3 dissociation experiments using the BD FACS™ Sortware version 1.0.1.654 software. Gating for eGFP was set using CMs from wild-type mice (Supplementary Fig. 12).

**Protein expression and purification.** BAG3[WT] and BAG3[P209L] cDNAs were cloned in pET-M11 (Novagen) and human HSPB8 and a human FLNC cDNA fragment corresponding to its Ig-like domains 19–21 into pET-28a (Novagen). His-tagged proteins were expressed and purified using Ni-NTA-agarose (Qiagen) as described by the manufacturer. An additional washing step with MgATP (2 mM ATP, 1 mM MgCl₂, 20 mM MOPS pH 7.2, 100 mM KCl) was added before elution. Rat HSPA8 was expressed in Sf9 insect cells following infection with a recombinant baculovirus carrying the corresponding coding region and subsequently purified on an ATP-sepharose (Sigma-Aldrich). IPV2[wt] and IPV2[P209L] encoding cDNA fragments (corresponding to aa 125–285 of human BAG3) and PPPY encoding cDNA fragments (corresponding to aa 537–670 of human TSC1) were subcloned into pMAL-c2 vector (New England Biolabs). MBP fusion proteins were expressed in *E. coli* BL21 (DE3) and purified on amylose resin (New England Biolabs) according to the manufacturer's protocol (with a MgATP washing step as above) before elution.

**In vitro binding studies and antibodies.** To investigate binding of BAG3[WT]/BAG3[P209L] to HSPA8, HSPB8 or FLNC, 1.5 μg purified protein was immobilized by a corresponding antibody on protein G-sepharose for 1 h at 4 °C and then incubated with 1.5 μM of putative interacting protein in 200 μl binding buffer for 2 h at 4 °C, followed by six washing steps with wash buffer and two washes with binding buffer. Retained proteins were eluted with glycine elution buffer (0.1 M glycine-HCl pH 3.5), precipitated in 15% trichloroacetic acid and analyzed by immunoblotting. The following antibodies were used for protein detection: anti-BAG3 (10599-1-AP, Proteintech Group, 1:1000), anti-His (MCA1396, Serotec, 1:1000), anti-HSPA8 (raised against purified rat HSPA8 following recombinant expression in insect cells, rabbit immunization and antibody purification were performed at BioGenes GmbH, 1:1000), anti-HSPB8 (#3059, Cell Signaling Technology, 1:1000) and anti-MBP (ab23903, Abcam, 1:1000). Uncropped blots and gels are presented in the Source Data.

**Partial trypsin digestion.** To detect structural variability between wild-type and mutant BAG3, 0.7 μM purified BAG3[WT] or BAG3[P209L] were incubated with 20 nM trypsin (T1426, Sigma) at 37 °C for 0, 0.5, 1, 2, 5 and 15 min. The reaction was stopped by addition of SDS sample buffer and immediate incubation at 95 °C for 10 min. Samples were analyzed by immunoblotting. Uncropped gels are presented in the Source Data.

**Heat denaturation**. To examine the thermal stability of BAG3[WT] and BAG3[P209L], 50 µl of purified protein (2 mg/ml) was incubated at 0, 60, or 70 °C for 15 min and denatured protein sedimented at $100,000 \times g$ for 30 min at 4 °C. Proteins in the supernatant and insoluble proteins were resuspended in an equivalent volume of SDS loading buffer. Supernatant and pelleted fractions were analyzed by SDS-PAGE, and stained gels were photographed using a Bio-Rad Chemidoc imaging system. Uncropped blots are presented in the Source data.

**Histology, immunofluorescence staining, and microscopy**. 10 µm thick cryosections were cut on a cryotome CM 3050S (Leica). Sirius red and Fast green staining was performed using standard histology protocols. For staining of fixated cells and heart sections the following antibodies were used (for 2 h at RT in 0.2% Triton X in PBS, 5% donkey serum or overnight at 4 °C in 0.2% Triton X in PBS, 5% donkey serum): α-actinin (1:200, Sigma-Aldrich), BAG3 (1:200, Proteintech, recognizing both human and mouse BAG3), desmin (1:100, DAKO), FLNC (1:1000, Biogene), SYNPO2 (1:200, Invitrogen), titin (1:50, T12[32]), cCasp3 (1:50, Cell Signaling Technology), CD45 (1:200, Merck Millipore), CD68 (1:100, eBioscience), vimentin (1:400, Merck Millipore), HSPA8 (1:200, Enzo Life Sciences), Endomucin (1:200, Santa Cruz), and HSPB8 (1:100, Abcam). After washing secondary antibodies conjugated to Cy2, Alexa Fluor 488, Cy3 or Alexa Fluor 647, respectively (all 1:400, Jackson ImmunoResearch), diluted in 1 µg ml$^{-1}$ Hoechst 33342 in PBS were applied for 1 h at room temperature. Sections were mounted with coverslips and imaged using an inverted fluorescence microscope (Axiovert 200, Carl Zeiss) with filters for DAPI, GFP, Cy3, and Alexa Fluor 647, and ×25, ×40 and DIC Plan Apochromat oil objectives, an ebx 75 light source and an AxioCam MRm digital camera. The editing of the pictures was performed with the Axiovision Rel. 4.8 Software (Zeiss).

**(Immuno)electron microscopy**. 4% PFA fixed heart tissue samples of 16-week-old mice were cut into longitudinal sections on a VT 1000S Leica vibratome and rinsed twice in PBS. For immuno-EM, samples were blocked in 20% NGS for 1 h and incubated with the respective primary antibodies in PBS supplemented with 5% normal goat serum overnight at 4 °C. The sections were then triple-washed with PBS and incubated with 1.4 nm gold-coupled secondary antibodies (Nanoprobes) overnight at 4 °C. After extensive washing, all sections were postfixed in 1% glutaraldehyde for 10 min and after rinsing, sections were reacted with HQ Silver kit (Nanoprobes). After treatment with OsO$_4$, samples were counterstained with uranyl acetate in 70% EtOH, dehydrated, and embedded in Durcupan resin (Fluka). Ultrathin sections were prepared with a Leica Ultracut S (Mannheim) and sections were adsorbed to glow-discharged Formvar-carbon-coated copper grids. Images were taken using a Zeiss LEO 910 electron microscope equipped with a TRS sharpeye CCD Camera (Troendle).

**Fluorescence recovery after photobleaching (FRAP) and data analysis**. Ventricular CMs isolated from αMHC-BAG3[WT]/BAG3[P209L]-eGFP mice were plated on 1% laminin-coated glass-bottom dishes and kept at 37 °C and 5% CO$_2$ during the entire experiment. FRAP experiments were performed and analyzed using a Cell Observer SD Spinning Disk Confocal Microscope (Carl Zeiss, Jena, Germany) equipped with an external 473 nm laser coupled via a scanner (UGA-40, Rapp OptoElectronic, Hamburg), using a Plan-Apochromat ×63/1.4 oil objective. For FRAP analyses single Z-disc regions of neighboring mature myofibrils were selected as regions of interest (ROI). In each cell 1–4 ROIs were analyzed. Photobleaching was performed with eight 1 ms pulses of the 473-nm laser (100 mW) at 100% intensity. Images were taken before and immediately after bleaching. Fluorescence recovery was subsequently monitored with an interval time of 0.5 s during the first 30 s and an interval of 1 s until the end of the experiment. The ImageJ package Fiji was used to determine fluorescence intensities of bleached and unbleached areas at each time point. Normalized FRAP curves were generated from raw data as previously described[33]. Briefly, to generate corrected FRAP curves, the intensity in the bleached ROI ($I_{frap}(t)$) and in the whole cell excluding the bleached area ($I_{total}(t)$) at each time point were initially subtracted by the corresponding background intensity ($I_{bg}(t)$). These corrected intensities were rescaled to pre-bleach intensities ($I_{frap-pre}$ and $I_{total-pre}$). This resulted in the following equation for calculating the normalized FRAP curve:

$$I_{frap-norm}(t) = \frac{I_{total-pre}}{I_{total}(t) - I_{bg}(t)} * \frac{I_{frap}(t) - I_{bg}(t)}{I_{frap-pre}}$$

Normalized fluorescence intensity versus time was plotted using Prism 4.0 (GraphPad Software). The exchange half-lives $t_{1/2}$ were calculated as follows:

$$t_{1/2} = \frac{\ln(2)}{K}$$

Mobile fractions ($M_f$) were calculated by the following equation:

$$M_f = \frac{F_{end} - F_{post}}{F_{pre} - F_{post}}$$

$F_{end}$ gives the fluorescence intensity at plateau level, $F_{post}$ is the fluorescence intensity immediately after bleaching and $F_{pre}$ is the fluorescence intensity at the beginning of the experiment. Results are given as mean ± SEM.

**Tissue RNA-Seq analysis**. For RNA-Seq experiments, DNA-free total RNA was isolated from total hearts from 6 CAG-BAG3[P209L] and 6 control mice using the RNeasy Kit (Qiagen) including on-column DNAse digestion. RNA quality was analyzed by an Agilent Bioanalyzer (Agilent). For library preparation the Trio RNA-Seq Library Preparation kit for mouse (TECAN) was used, starting with 50 ng of total RNA. Thirteen PCR cycles were used for library amplification and libraries with an average fragment size of 380 bp were sequenced on a NextSeq 500 in paired-end mode (75 bp, Illumina). For bioinformatics analysis, we used the Galaxy platform (Freiburg Galaxy Project[34]). RNA sequencing reads were mapped using RNA STAR[35] followed by counting reads per gene by using featureCounts[36]. Differentially expressed genes were identified by DESeq2[37]. For data visualization, normalization and cluster analysis heatmap2 and Volcano plot (Freiburg Galaxy Project[34]) was used. Gene ontology analysis was performed by ClueGO (two-sided hypergeometric test) by using the KEGG pathway and GO-term databases with a significance interval for pathways of $p < 0.05$. $p$-values were corrected for multiple-testing by the Bonferroni step down method.

**Single-cell RNA-Seq analysis**. Single CMs from Langendorff dissociated mouse hearts ($n = 2$ control hearts and $n = 2$ CAG-BAG3[P209L] hearts) were manually picked into each well of 96-well plates which were filled with 9.5 µl of lysis buffer (0.25 µl RNase Inhibitor, 0.05 µl 10% Triton, 2.5 µl 10 mM dNTP, 6.7 µl H$_2$O). Single-cell mRNA sequencing procedure was carried out at the Stanford Functional Genomics Facility by following the SMART-seq2 protocol[38]. Briefly, the single CMs were lysed, and their mRNA was reverse transcribed into cDNA and pre-amplified. After a cleanup procedure, the cDNA was quantified and used to make single-cell libraries following the manufactural protocol in Illumina Nextera DNA library kit. The single-cell libraries were further sequenced on an Illumina HiSeq 4000 platform using $2 \times 75$ bp mode.

The sequencing reads for each cell were mapped to a reference database with the mouse genome and human BAG3 and eGFP transcript sequences using STAR2.7.0f with the default parameters, and gene expression counts were calculated using STAR "–quantMode GeneCounts" parameter. Further data analysis was carried out in the Seurat V3.1 using default parameters[39]. Briefly, the single cells were filtered based on their total expressed genes, normalized using a global-scaling normalization method "LogNormalize", and scaled with the "ScaleData" function. After further linear and non-linear dimensional reductions, the expression ratio of transgenic human BAG3 to mouse endogenous Bag3 was plotted using the VlnPlot feature. Differential expression analysis was performed as a cluster analysis from the single-cell RNA-Seq data. Clusters of single cells from CAG-BAG3[P209L] mice ($n = 80$ bright CMs from 2 mice) and control mice ($n = 33$ CMs from 2 mice) were pooled and analyzed for differentially expressed genes by using FindMarkers function in Seurat V3.1.

**Proteome sample preparation and data acquisition**. Heart samples from the mouse were harvested, weighed, and frozen in liquid N2 until use. For proteome analysis about 100 mg of heart sample was homogenized using a TissueLyser LT (Qiagen) in 1 ml of 50 mM Tris buffer (pH 6.8) containing 8 M Urea, 2% SDS, 1.5% Triton-X100, 1 mM DTT, 1 mM PMSF and 1:100 (v/v) protease inhibitor cocktail. The protein concentration was measured using Pierce BCA protein assay kit (Thermo Scientific) and 100 µg proteome was reduced using 10 mM DTT at 37 °C for 30 min followed by carbamidomethylation using 50 mM chloroacetamide at room temperature (RT) in dark for another 30 min. The reaction was quenched with 50 mM DTT at RT for 20 min. Proteins were purified using paramagnetic SP3 beads, and eluted with the digestion buffer containing 50 mM HEPES pH 7.4, and 5 mM CaCl$_2$. Trypsin (Serva) was added to the eluted proteome in a 1:100 (w/w) ratio and incubated in a shaker at 37 °C for 18 h. Digested peptides were de-salted by reverse-phase solid phase extraction using self-packed C18 STAGE tips, before analysis with a nano-LC system (Dionex NCS-3500 RS) operated in two column-setup (PharmaFluidics µPAC C18 trap column and a 50 cm PharmaFluidics µPAC C18 analytical column). 500 ng peptide were separated at a flow rate of 600 nl/min with a binary gradient from 2 to 32% eluent B (A, 0.1% formic acid in HPLC-grade water; B, 0.1% formic acid in acetonitrile) and introduced into a high-resolution Q-TOF mass spectrometer (Impact-II, Bruker) using a CaptiveSpray ion source. The HyStar Software (v3.2, Bruker Daltonics) was used for data acquisition in line mode in a mass range from 200 to 1750 $m/z$ with an acquisition rate of 4 Hz for MS1 spectra and the top 17 most intense ions were selected for fragmentation. A dynamic exclusion window of precursors selected within 30 s was applied unless the signal to noise ratio improved more than 3-fold. Fragmentation spectra were acquired between 5 Hz for low-intensity precursor ions (>500 cts) and 20 Hz for high intensity (> 5k cts) ions, each with stepped parameters, each with 50% of the acquisition time dedicated for each precursor: 100 µs transfer time, 7 eV collision energy and a collision RF of 1500 Vpp followed by 100 µs transfer time, 9 eV collision energy and a collision RF of 1700 Vpp.

**Proteomics data analysis**. Spectrum to sequence matching was performed with MaxQuant version 1.6.6.0 and 1.6.10.43[40] for data acquired from 2 week-old ($n = 6$ control and 6 CAG-BAG3[P209L] hearts) and 5-week-old mice ($n = 6$ control and 6 CAG-BAG3[P209L] hearts), respectively, using standard settings for Bruker Q-TOF instruments. The UniProt Mus musculus protein database with appended common

contaminants listed in MaxQuant, the human BAG3[P209L]-GFP sequence, and reverse-decoy sequences were used for target-decoy database searches with an FDR of 0.01 at the PSM and protein level. Label-free quantification (LFQ) and the "match between runs" features were enabled. Trypsin was set as digestion enzyme, oxidation (M), and acetylation (protein N-term) were set as variable modifications while carbamidomethylation of cysteine was set as fixed modification. The LFQ data were analyzed using Perseus[40] version 1.6.10.0, filtering for proteins quantified in at least four biological replicates/animals in at least one of the two genotypes, followed by imputation of missing values using Perseus standard settings and determination of significant changes with a two-sample t-test with Benjamini–Hochberg FDR < 0.05 to correct for multiple hypothesis testing.

**Immunoblot analysis of mouse hearts**. For protein isolation, hearts were mechanically homogenized in urea buffer (8 M urea, 2% SDS, 1.5% Igepal, 0.05 M Tris-HCL pH 6.8 in aqua bidest) with protease inhibitors (Roche). Protein concentration was determined using the Pierce BCA Protein Assay Kit (Thermo Fischer Scientific). SDS-PAGE was performed in a 12.5% polyacrylamide gel. Proteins were then transferred to a nitrocellulose membrane using the semi-dry blotting method. The following primary antibodies were used: Anti-alpha B Crystallin (1:5000, Enzo Life Science), anti-HSPB7 (1:1000, Abcam), anti-HSPB8 (1:1000, Abcam) anti-LC3B (1:1000, Thermo Fisher Scientific), anti-p62 (1:1000, Progen), anti-Bag3 (1:5000, Proteintech), anti-GAPDH (1:1000, Calbiochem), anti-mCherry (1:5000, Novus), beta-actin (1:2000, Invitrogen). A peroxidase-coupled anti-mouse antibody (1:10,000, Jackson ImmunoResearch), a Cy3-coupled anti-rabbit antibody (1:3000, Jackson ImmunoResearch), and a Alexa Fluor 647-coupled anti-Rabbit antibody (1:3000, Jackson ImmunoResearch) were used as secondary antibodies. Chemiluminescent signals were detected by using the SuperSignal[TM] West Pico Plus Substrate (Thermo Fischer Scientific) and a ChemiDocTM MP Imaging System (BioRad) and fluorescence signals directly with the ChemiDocTM MP Imaging System (BioRad). Integrated intensities of protein bands were quantified using the Fiji (ImageJ) version 1.47n or the Image Lab version 5.2.1 software. Uncropped blots are presented in the Source data.

**Fractionation and immunoblot analysis of mouse hearts**. Frozen hearts were homogenized in 2 ml safe-lock tubes (Eppendorf) with 7 mm stainless steel beads (Qiagen, 69990) in 16 µl/mg tissue of low salt buffer (30 mM phosphate buffer pH 6.8, 100 mM KCl, 5 mM EDTA, 5 mM EGTA, 1 mM DTT and protease inhibitors (Sigma-Aldrich) for 5 min at 50 Hz in a precooled Tissue Lyser LT (Qiagen). After sonication on ice using an ultrasonic processor (UP100, Hielscher) at a cycle of 0.9 and an amplitude of 100% for at least 10 bursts, the insoluble fraction was pelleted by centrifugation (16,000 × g for 15 min). The supernatant was collected, and the pellet was solubilized in 16 µl urea buffer (100 mM Tris pH 6.8, 2 M thiourea, 7 M urea, 5 mM EGTA, 5 mM EDTA, 1 mM DTT and protease inhibitors) per mg pellet using the Tissue Lyser (2 min, 50 Hz) as described above, followed by incubation for 15 min at 37 °C. Preheated 5× concentrated SDS sample buffer was added to the supernatant and the solubilized pellet to a final concentration of 2-fold and samples were incubated for 5 min at 55 °C. All extracts were analyzed by polyacrylamide gel electrophoresis. Gels were stained with Coomassie Brilliant Blue R-250 Dye and the relative total protein concentration was measured using a LI-COR Odyssey Infrared Imaging System (LI-COR Biosciences). For comparative quantitative blotting, identical total protein amounts were separated on polyacrylamide gels, and proteins were transferred onto PVDF membranes using a Transblot SD apparatus (Biorad). Ponceau Red stain confirmed efficient transfer. Membranes were incubated with rabbit polyclonal anti-BAG3 antibody (Proteintech 1:5000) and IRDye-800-conjugated goat anti-rabbit secondary antibody (LI-COR Biosciences, 1:10,000), and analyzed using a LI-COR Odyssey Infrared Imaging System (LI-COR Biosciences). Integrated intensities of protein bands were quantified using the Odyssey Infrared Imaging Software v. 3.0 (LI-COR Biosciences). The efficiency of the fractionation was tested by staining with anti-pan MyHC (clone MF20, 1:100), recognizing all myosin heavy chain isoforms and anti-GAPDH (1:1000) followed by IRDye-800-conjugated goat anti-mouse secondary antibody (LI-COR Biosciences, 1:10,000). This confirmed the presence of myosin heavy chains only in the insoluble fraction and GAPDH only in the supernatant. Uncropped blots and gels are presented in the Source data.

**Injection of AAV/rh10**. For a gene therapy approach AAV/rh10-mCherry-U6-h-BAG3-shRNA (RNAi) and AAV/rh10-mCherry-U6-scrmb-shRNA (control) were used (Vector BioLabs). Both viruses contain a CMV-driven mCherry cDNA and either a U6-promoter-driven shRNA against human BAG3 (target sequence CTTGAACAGAAAGCCATTGAT) or a randomly scrambled shRNA. A total volume of 100 µl PBS containing 2 × 10¹² virus particles of either RNAi or control AAV/rh10 was injected into the left jugular vein of P15 CAG-BAG3[P209L] mice. Mice were dissected 3 weeks later (P37) and analyzed.

**Statistics**. Data are presented as means ± SEM or means ± SD. Unpaired t-test (two-sided) or one-way ANOVA (one-sided) with Bonferroni's multiple comparison test was used to test for significance. P-values <0.05 were considered statistically significant (GraphPad Prism V9.0.1). All measurements were taken from distinct samples. The minimal sample size for all mouse studies was determined by a power analysis using PS Power and Sample Size Calculations program V3.1.6 from Vanderbilt University (http://biostat.mc.vanderbilt.edu/PowerSampleSize) with the following assumptions: We are planning a study of a continuous response variable from independent control and experimental subjects with 1 control per experimental subject. In a previous study, the response within each subject group was normally distributed with standard deviation 0.1. If the true difference in the experimental and control means is 0.2, we will need to study 5 experimental subjects and 5 control subjects to be able to reject the null hypothesis that the population means of the experimental and control groups are equal with probability (power) 0.8. The Type I error probability associated with this test of this null hypothesis is 0.05.

**Study approval**. All applicable international, national, and/or institutional guidelines for the care and use of animals were followed. All procedures performed in studies involving animals were following the ethical standards of the institution at which the studies were conducted and were approved by the responsible governmental animal care and use office, the Landesamt für Natur, Umwelt und Verbraucherschutz, LANUV (84-02.04.2012.A146 and 81-02.04.2019.A062).

**Housing and husbandry of mice**. All mice are housed under Specific-Pathogen-free (SPF) conditions in individually ventilated cages in the animal facility of the University Clinic Bonn. Our surveillance program follows the Federation of European Laboratory Animal Science Associations (FELASA) guidelines. A maximum of 5 mice are housed in Type 2 L cages on a 12/12 h light/dark cycle and mice have ad libitum access to food and water. The enclosures for laboratory rodents have a standardized air temperature of 22 °C, 50–70% humidity and up to 15-fold air exchange. Bedding material is added in all mating cages and small houses and toys are routinely added to enrich the environment of the mice. Access to the facilities is restricted to qualified personnel that received appropriate training in animal handling and experimentation. The facility has appointed a certified veterinarian, who regularly monitors the sanitary status of the facilities and the health status of all the animals kept.

**Reporting summary**. Further information on research design is available in the Nature Research Reporting Summary linked to this article.

## Data availability

MS data have been deposited to the ProteomeXchange Consortium via the PRIDE partner repository[41] with the project accession number PXD021165. Single cell RNA-Seq data have been deposited to the GEO database with the GEO accession number GSE166862. Tissue RNA-Seq data have been deposited to the SRA database with the BioProject accession number PRJNA700583. Source data are provided with this paper. All other source data are provided upon request.

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

## Acknowledgements

We thank S. Grünberg (Bonn, Germany), P. Freitag (Bonn, Germany), D. Korzus (Bonn, Germany), J. Schmidt (Bonn, Germany), and K. Bois (Bonn, Germany) for excellent technical assistance. We also would like to acknowledge A. Nagy (Toronto, Canada) and M. Gertsenstein (Toronto, Canada) for providing the G4 mouse ESC line. This work was supported by the German Research Foundation (FOR1228 to D.O.F. and FOR1352 to B. K.F., D.O.F., and J.H. FOR2743 to J.H., P.F.H., D.O.F., M.H.; Collaborative Research Centre 1425, P14 to B.K.F. and W.R., P03 to L.H., and P05 to A.L.) and the Seventh Framework Program for Research and Technological Development of the EU (MUZIC; D.O.F.). The Galaxy server that was used for some calculations is in part funded by Collaborative Research Centre 992 Medical Epigenetics (DFG grant SFB 992/1 2012) and German Federal Ministry of Education and Research (BMBF grants 031 A538A/A538C RBC, 031L0101B/031L0101C de.NBI-epi, 031L0106 de.STAIR (de.NBI)).

## Author contributions

K.K., A.O., and K.G.R. performed molecular biology, cell culture, imaging, and immunohistochemistry experiments and data analysis. W.A.L. and A.U. designed, performed, and analyzed (immuno)electron microscopy experiments. C.G. was involved in molecular biology experiments and data analysis. D.O.F., J.S., and P.F.M.vdV. performed FRAP, heat denaturation experiments, and immunoblotting. W.R. and W.B. were involved in small animal surgery and treadmill exercise experiments. M.K. and P.F.H. performed proteomics analysis. J.D. and J.H. designed and performed immunoprecipitation, in vitro binding studies, and partial trypsin digestion experiments. A.L. and L.H. performed echocardiography analysis. M.H. and S.T. were involved in the generation of transgenic mice and M.H. performed RNA-Seq analysis. S.M.W. and G.L. performed single-cell RNA-Seq experiments and analysis. M.H. and B.K.F. designed the study and wrote the manuscript (together with D.O.F.).

## Funding

## Competing interests

The authors declare no competing interests.
