## [Peer Review File · Nature Communications]

REVIEWER COMMENTS

Reviewer #1 (Remarks to the Author):

Patients who harbor a single nucleotide polymorphism in exon 3 of BAG3 that results in a substitution of a leucine for a proline at amino acid position 209 (P209L) have childhood progressive and severe muscle weakness, respiratory insufficiency, and a restrictive/hypertrophic cardiomyopathy, and elevation of serum creatine kinase levels. In this report, the authors generated humanized transgenic mouse models expressing human BAG3P209L in mice caused Z disc disintegration, formation of protein aggregates, fibrosis and an early-onset restrictive cardiomyopathy with increased mortality. Interestingly the mice had the same phenotype as humans. The authors then tested a gene therapy approach utilizing expression of shRNA against hBAG3 in cardiac muscle via AAV and demonstrate that this halted and partially reversed major phenotypic features of the disease.

I have a number of comments for the authors;

1. why did the authors use alpha MHC as their cardiac restricted promoter. As the mice develop cardiomyopathy or age they decrease in alpha MHC and increase in beta MHC and that may dilute the effects of the defective gene
2. There are significant differences between this model and the zebrafish model created a number of years ago. Studies in zebrafish to identify the mechanisms of action of P209L BAG3-P209L localized to the Z disk and were associated with the accumulation of small granular protein aggregates containing wild-type BAG3 within the striations. However, the formation of these protein aggregates was not attributable to impaired autophagy.
3. BAG3 mutation affects the beta receptor G protein pathway in cardiac tissues. Have the authors examined the beta receptor pathway in their model.
4. The gene therapy study is surprising. sh against BAG3 seems to delay the phenotype in the transgenic mice, however other studies have shown deleterious effects of sh BAG3 in cardiac cells.

Reviewer #2 (Remarks to the Author):

The study described in this manuscript was well designed and carefully executed. Quantifying transgene penetrance in CMs turned out to be a necessary step before phenotyping Tg mouse lines. The resulting Tg CAG-hBAG3^{P209L} mouse line mimicked key cardiac phenotypes of human disease. However, minor issues could be addressed to improve the clarity and readability of this manuscript. See detailed comments below.

Issues to be addressed:

1) The proteomics protocols employed are common practice in the field; there are a few items requiring your attention: the soluble proteins and protein aggregates are not separated. During protein extraction, the sequestered proteins would be dissolved by the 8M urea buffer and then digested together with soluble proteins for the label-free proteomics quantification. This approach will not give you the full information you might have gotten if you separate them. The unknown distribution of individual proteins between the sequestered (w/o function) pool and soluble (w/ function) pool might complicate functional annotation. Would the authors consider including the lists of significantly regulated genes and proteins identified by RNA-seq and proteomics analyses in supplemental data? These would provide essential details of volcano plots and heatmap.

2) In Fig 6A, insoluble pellet fractions were isolated by differential centrifugation of cardiac protein extracts from both CAG-hBAG3^{P209L} mice and CAG-hBAG3^{WT} mice. The protein aggregates were implied to be the predominant component of these insoluble fractions. Were the pellets collected in largely reduced yields from the CAG-hBAG3^{WT} mouse hearts when protein aggregates were not observed in hearts of the CAG-hBAG3^{WT} mice at 5 wks of age (Suppl. Fig. 10C-F)?

In Fig, 6B. It is not clear how the “relative amounts” of hBAG3 protein and endogenous mBAG3 protein were defined. What was used to normalize the WB band intensities as relative ratios of protein expression?

A detailed protocol of differential fractionation (or citation) is missing. Some quality control validation would be helpful to demonstrate an efficient separation of protein aggregates from other insoluble proteins (e.g., cytoskeleton, contractile proteins).

3) Suppl. Fig. 9F, the P209L panel showed multiple IB bands at lower M.W. positions. Are they fragments of hBAG3^{P209L} protein? If so, these protein fragments exist in purified

hBAG3^{P209L} samples. What could be the possible cause for generating these fragments? Were these low M.W. bands included in quantification?

4) On Page 24, L521-523: the Statement "Mice ... did not display any obvious phenotype despite the strong hBAG3^{WT}-eGFP expression" should be supported by Suppl. Fig. S10 instead of S9.

5) Fig. 7B. The last lane of WB shows almost no mCherry expression; it is unclear whether shRNA was effective in this sample. It would be more informative if the loading controls were included.

6) The manuscript could use another round of editing to improve clarity. Multiple typos, inconsistent abbreviations, and mislabeling were noticed. Several examples are listed below,

- Both "αMHC-BAG3^{WT}-eGFP" and "αMHC-BAG3^{WT}" were used representing the same mouse line; the same issue for "αMHC-BAG3^{P209L}-eGFP" and "αMHC-BAG3^{P209L}".

- Fig. 1B-D legend "Langendorff-isolated CMs from 10 weeks old WT, αMHC-BAG3^{WT}, and αMHC-BAG3^{P209L} mice stained for α-actinin (white) displaying Z-disc localization of BAG3^{WT} (insert)": Please consider separating legends for (B) and for (C-D); WT strain was used in (C-D) only; does "insert" mean "zoomed region"?

- Fig. 1B legend: Details describing the figure would be helpful, "While BAG3^{WT} (green, *top panel*) formed a grid-like structure (zoomed region, *top panel*), BAG3^{P209L} (green, *bottom panel*) was found to form aggregates of different sizes (arrows, *bottom panel*)."

- Fig. 1F. "Immunogold stainings (black dots)" were conducted only with αMHC-BAG3^{P209L} mouse heart. This image shows that sarcomeric disarray was observed specifically in eGFP-positive but not in eGFP-negative CMs of the same heart.

- According to main text (P6, Lines 105-108), the eGFP+ CMs were isolated from both αMHC-BAG3^{WT} and αMHC-BAG3^{P209L} Tg mouse hearts, respectively. Immunoblotting was conducted using the isolated CMs (Suppl. Fig. 1C-D). The current legend of Suppl. Fig. 1C-D mislabelled these samples as being obtained from the whole heart.

Additional Points for Your Considerations:

- Line 68: Remove comma after disruption.

- Line 69: Pick a simpler phrase for "in congruence with". Italicize "in vitro"

- Line 82: Is there a simpler word than "recapitulate", here and elsewhere? Perhaps "mimic" or "imitate"?

- Line 94: Should there be a hyphen in "10 weeks" if there is one after "3"?

- Line 96: It could benefit readability to be consistent with using an Oxford comma or not (putting a comma before "and" in a list). Elsewhere, it was used, but here it wasn't.
- Line 127: "hBAG3 expression" would read better than "expression of hBAG3". This rule can be applied elsewhere when deemed helpful.
- Line 136: Add a hyphen between "eGFP" and "marked" if the structures are marked with eGFP.
- Line 137: Add a comma after "antibody" before "recognizing" or clarify what the second part (with "recognizing") is referring to.
- Line 149: Do "neither presented" instead of "did neither present".
- Line 161: "Bred to homozygosity with PGK-Cre" may read better.
- Line 163: Do "mild" and "variable" contradict each other? Or is there a wide enough range for "mild" to still be variable?
- Line 168: Add a comma before "displaying". As is, it could read that the littermates displayed growth retardation, not that the CAG-BAG3 mice displayed growth retardation.
- Line 170: "euthanized" in lieu of "sacrificed".
- Line 172: Choose a consistent hyphenation pattern.
- Line 177: "on hearts" reads better than "of hearts". Or it would be better to put "hearts" at the end of the sentence ("fluorescence on 1 to 5 week old CAG-BAG3P209L mices' hearts.").
- Line 180: If scientifically accurate, "Transgene expression in hearts" would be better and more readable than "The expression of the transgene in hearts".
- Line 189: Add "that" after "found".
- Line 198: "caused by collagen deposits replacing apoptotic cells" reads easier. It removes a "by" so it is more straightforward.
- Line 258: Change to "indicative of"

Reviewer #3 (Remarks to the Author):

I have carried out a technical review of the RNAseq in this manuscript.

The authors generated a transgenic mouse line with conditional expression of hBAG3P209L-eGFP under the control of the CAG-promoter (CAG-BAG3P209L), which was found to be exclusively expressed in CMs by immunofluorescence staining.

They performed single-cell RNA-seq on the control and CAG-BAG3P209L mice at 2 weeks of age, to determine the ratio of hBAG3 to mouse Bag3 expression in eGFP+ CMs. Subsequently, they also performed bulk RNA-seq on 50-70 CMs, and tissue RNA-seq to compare CAG-BAG3P209L to control mice.

1. It is not mentioned how many cells were included in single-cell RNA-seq.
2. In methods (page 30), it is mentioned that the sequencing reads were mapped to the mouse genome. It is unclear how the ratio of human BAG3 to mouse Bag3 was determined if the reads were only mapped to the mouse genome. It is also unclear how the ratio was determined (was it based on the total number of reads mapped? Was it normalized by the gene length, etc?) since methods only state that the expression ratio was plotted using VlnPlot feature.
3. On page 12, "These experiments showed that the ratio of hBAG3 to mouse Bag3 was in the range of 0.5 to 2-fold, ... (Suppl. Fig. 7A)". In Supplementary Figure 7A, the range of y-axis is set from log₂ value of 1, which is equivalent to 2 fold ratio. This would mean hBAG3 to mouse Bag3 ratio was more than 2 fold in all cells, even in the control mice without the transgene. This also does not correspond to the range (0.5-2 fold) quoted in the main text. Moreover, it is unclear why some cells showed a high ratio of hBAG3/mBag3 in control mice.
4. List of differentially expressed genes should be reported as a supplementary data.
5. Significance of GO-analysis should be reported as a supplementary data (size of gene sets, level of significance, list of genes included, etc.)
6. Authors reported 35 downregulated genes from bulk RNA-seq of a pool of 50-70 individual CMs, while 1854 downregulated genes were reported from the tissue RNA-seq. From how many mice were the 50-70 CMs obtained? How many mice were included in tissue RNA-seq? The difference between the number of downregulated genes is striking and maybe many of the downregulated genes detected in tissue RNA-seq arises from the differences between mice. Reporting the number of replicates included in tissue RNA-seq would minimize confusion around this.
7. Minor spelling errors (e.g. "Volcano blot" in page 45)
8. The color scale of Figure 5B indicates "log₂(fold of mean)". Are the values in the heatmap fold changes, or actual expression values? Out of 3804 differentially expressed genes, how were the genes shown in Figure 5B selected? It should also be discussed clearly

Reviewer #4 (Remarks to the Author):

Kimura et al – referee report Nat Comm

In their manuscript, Kimura et al report on the phenotypes of a humanized transgenic BAG3P209L mouse models that they generated. The mice show massive fibrosis resulting in a severe, early-onset restrictive cardiomyopathy with increased mortality and histological analyses shows Z-disc disintegration and formation of protein aggregates, all similar to what has been observed in patients.

Previous attempts by others had failed to reveal such a phenotype, which -as the authors suggest- may have been due to too low expression levels. In fact, the authors confirm this idea and now show that mice with lower expression do already show protein aggregation and mild morphological changes, but do not (yet) present signs of cardiomyopathy, compromised heart function and early mortality. A second model, however, in which higher expression levels were achieved, shows not only more (rapid) aggregation and more morphological damage to sarcomeric structures, but also a clear functional phenotype. Besides early onset skeletal muscle weakness, these mice displayed clear signs of cardiomyopathy, and at 5 weeks of age suffered from severe heart failure or died prematurely. This is totally in line with the progressive character of this disease in humans. It also supports the concept that progressive protein aggregation drives this disease.

The data on this part are clear, novel, and valuable as these mice may hence be a important tool for evaluating therapeutic strategies for this disease.

Even more so, as BAG3 levels are upregulated by different forms of stress, the current data also have relevance to preventive measures in these patients (i.e. to avoid physical stress), which is a message that could be more strongly conveyed.

Using RNA-Seq analyses in 5-week-old mice, the authors next show a number of differentially expressed genes, consistent with the degenerative features in the cardiomyocytes (lower expression of genes involved in energy metabolism, oxidative phosphorylation and cardiac muscle contraction) and the presence of a proteotoxic stress (upregulation of genes various (protein) quality control pathways). Proteomic analysis nicely corroborated these findings. Whilst interesting as observation

and the data are well presented, these data add little information in terms of cause-consequence relations, except for the fact that many of the changes found involve BAG3-interactors. The latter indeed would be consistent with (but by themselves provide no proof for) suggestions from previously published data on a BAG3-driven protein aggregation process that has triggered a rather broad protein stress response (Hsp's, autophagy), maybe as an attempt to restore protein homeostasis.

More revealing in terms of mechanisms are their data to show that the mutant human BAG3P209L associated normally with its partners HSPA8 and HSPB8 (i.e similar to BAG3WT) but accumulates in insoluble protein aggregates (as revealed by both biochemical and immunofluorescent, FRAP analyses) in which also the endogenous, wildtype mouse BAG3 is sequestered. As correctly referenced by the authors, this is completely consistent with previously published work in cell models. So, whereas these data do not bring in a novel concept, they are important as an in vivo validation of the published cell biological work and supports the idea that also at the organismal levels the BAG3P209-related disease progression can be linked to a progressive collapse of protein homeostasis.

Finally, the authors show how gene therapy using AAV-mediated knockdown of hBAG3 strongly mitigates the disease phenotype. Whereas well-executed and showing the proof-of-concept that AAV strategies can lead to a knockdown of a transgenic human BAG3P209L in mice and reverse its toxicity (which is not really a surprise), I find their conclusion that this demonstrates 'that an AAV-based gene therapy is a feasible and promising therapeutic strategy to halt or even reverse the progression of this severe myofibrillar myopathy' (discussion) premature. The crux of the problem for such a strategy lies in the none-specificity of such an approach that also would target the wildtype allele. This likely will result in major toxicity issues, given the importance of BAG3 for cellular physiology. I am aware of and do appreciate the fact that the authors themselves also write that shRNA specific to the mRNA carrying the point mutation has to be identified. However, this seems easier said than done and as is true for the suggested alternative of CRISPR-based gene-editing tools. So, here their 'promising' statement should be tempered.

Minor comments

Introduction:

- To my knowledge BAG3 does NOT balance protein synthesis in response to stress but it does alters/increases protein degradation in response to stress, but also not exclusively upon mechanical stress (although this could be a major physiologically relevant stress). This should be revised.
- There is no evidence to show that BAG3 itself recognizes mechanically unfolded and damaged forms of FLNC. Rather, data suggest that in conjunction Hsp70 proteins, it targets Hsp70 recognized

client proteins to autophagy and does so under conditions of proteasomal overload, conditions by which it is upregulated. This is also why CASA as terminology for this process is confusing. Moreover, albeit that HSPB8 is a preferred small HSP partner of BAG3, HSPB8 is dispensable for the BAG3dependent autophagosomal targeting of clients (see the work of Carra and co-workers on this matter).

Results

- ...the sarcomeric protein titin, which is not under control of BAG3-mediated proteostasis....

Proteostasis refers to protein homeostasis and not protein quality control only.

Please rephrase to : ...the sarcomeric protein titin, which is not under control of BAG3-dependent protein quality control.....

- It is very interesting though that titin is not affected, whilst FLNC and desmin are:

To me this does not imply that “diverse Z-disc components are subjected to distinct aggregation pathways”, but rather that they may depend on different protein quality control systems for their integrity or that titin is more stable/ less affected by the aggregation cascade initiated by BAG3P209L. Besides my request to rephrase this sentence accordingly, I wonder whether the authors could speculate why this could be different.

Point-by-point response to Reviewer Comments

Reviewer #1 (Remarks to the Author):

Patients who harbor a single nucleotide polymorphism in exon 3 of BAG3 that results in a substitution of a leucine for a proline at amino acid position 209 (P209L) have childhood progressive and severe muscle weakness, respiratory insufficiency, and a restrictive/hypertrophic cardiomyopathy, and elevation of serum creatine kinase levels. In this report, the authors generated humanized transgenic mouse models expressing human BAG3^{P209L} in mice caused Z disc disintegration, formation of protein aggregates, fibrosis and an early-onset restrictive cardiomyopathy with increased mortality. Interestingly the mice had the same phenotype as humans. The authors then tested a gene therapy approach utilizing expression of shRNA against hBAG3 in cardiac muscle via AAV and demonstrate that this halted and partially reversed major phenotypic features of the disease. I have a number of comments for the authors;

1. why did the authors use alpha MHC as their cardiac restricted promoter. As the mice develop cardiomyopathy orage they decrease in aplha MHC and increase ib beta MHC and that may dilute the effects of the defective gene

Answer: Alpha MHC expression in the mouse heart strongly increases after birth, whereas beta MHC is downregulated. Therefore, alpha MHC-driven expression of a transgene is the preferred strategy by most groups to achieve its CM-specific overexpression. This is also true for cardiomyopathy models, in which only a relatively modest increase in the beta MHC/alpha MHC-ratio can be observed. In our case, the recombination rate was very low and therefore did not result in a pronounced cardiac phenotype. This was the reason, as reported in the ms, why we generated the second model that phenocopies key features of the human disease.

2. There are significant differences between this model and the zebrafish model created a number of years ago. Studies in zebrafish to identify the mechanisms of action of P209L BAG3-P209L localized to the Z disk and were associated with the accumulation of small granular protein aggregates containing wild-type BAG3 within the striations. However, the formation of these protein aggregates was not attributable to impaired autophagy.

Answer: Although there are some differences, the zebrafish models generated so far (Ruparella et al., 2014; Ruparella et al., 2020) generally are in accordance with our mouse model. In the first report (Ruparella et al., 2014) bag3 was knocked down using morpholinos followed by human BAG3^{P209L}-eGFP overexpression. In this fish model BAG3^{P209L} aggregates formed and thereby reduced the pool of functioning BAG3. Although impaired autophagy was excluded to be the cause of the formation of protein aggregates it could not be excluded that autophagy was impaired as a consequence of BAG3^{P209L} aggregation. The second study (Ruparella et al., 2020) demonstrated that human Bag3^{P209L}-eGFP expression in a bag3^{-/-} background can fulfill the function of endogenous bag3. It was also shown that it tends to aggregate and that loss of BAG3 in aggregates leads to impaired macroautophagy/autophagy. This perfectly matches the data and working hypothesis in our mouse models, as we also see mouse Bag3 in aggregates together with hBAG3^{P209L}-eGFP and have

therefore concluded that the absence of Bag3 leads to impaired autophagy. As a secondary effect, loss of function of Bag3 impairs the CASA complex and thus part of the protein quality control system, leading to even more protein accumulation. This is underscored by the finding of increased expression of autophagy-related proteins (transcriptomics) and their accumulation (proteomics), suggesting impaired autophagy.

To point this out and to make our hypothesis clearer we have reworded parts of the revised discussion sections (page 21, lines 458-459).

3. BAG3 mutation affects the beta receptor G protein pathway in cardiac tissues. Have the authors examined the beta receptor pathway in their model. (is this the case, indirect or direct (?):

Answer: We thank the reviewer for bringing up this interesting point. Neither transcriptomics nor proteomics indicate in our model prominent beta-receptor involvement. To date we have not explored this aspect in more detail, but will address this in future experimental work.

4. The gene therapy stud is surprising. sh against BAG3 seems to delay the phenotype in the transgenic mice, however other studies have shown deleterious effects of sh BAG3 in cardiac cells.

Answer: In our proof-of-concept approach, only human BAG3^{P209L} is downregulated, whereas endogenous mouse Bag3 is not affected and thus can fulfill its function. This explains the discrepancy to the published knockdown experiments in zebrafish (Ruparelia et al., 2014) or in mouse left ventricular cardiomyocytes (Feldman et al., 2016). We fully agree with the reviewer that knockdown of both the BAG3^{P209L} and endogenous Bag3 would most likely lead to toxic effects, as mentioned in the Discussion section. We are also aware of the challenges in designing a point mutation-specific shRNA (Schwarz et al., 2006), which may have to be explored in a future study. Alternatively, either micro-RNAs (Acunzo et al., 2017) or a CRISPR/Cas9-based approach such as prime editing could be tested. We have added the potential limitations and drawbacks of these approaches to the revised version of the Discussion section (page 22, lines 475-481).

Ruparelia AA, Oorschot V, Vaz R, Ramm G, and Bryson-Richardson RJ. Zebrafish models of BAG3 myofibrillar myopathy suggest a toxic gain of function leading to BAG3 insufficiency. *Acta Neuropathol.* 2014;128(6):821-33.

Schwarz DS, Ding H, Kennington L, Moore JT, Schelter J, Burchard J, et al. Designing siRNA that distinguish between genes that differ by a single nucleotide. *PLoS Genet.* 2006;2(9):e140.

Acunzo M, Romano G, Nigita G, Veneziano D, Fattore L, Laganà A, et al. Selective targeting of point-mutated KRAS through artificial microRNAs. *Proc Natl Acad Sci U S A.* 2017;114(21):E4203-e12.

Reviewer #2 (Remarks to the Author):

The study described in this manuscript was well designed and carefully executed. Quantifying transgene penetrance in CMs turned out to be a necessary step before phenotyping Tg mouse lines. The resulting Tg CAG-hBAG3^{P209L} mouse line mimicked key cardiac phenotypes of human disease. However, minor issues could be addressed to improve the clarity and readability of this manuscript. See detailed comments below.

Answer: We thank the reviewer for his/her positive comments on our work and the constructive suggestions to further improve the presentation of the data.

Issues to be addressed:

1) The proteomics protocols employed are common practice in the field; there are a few items requiring your attention: the soluble proteins and protein aggregates are not separated. During protein extraction, the sequestered proteins would be dissolved by the 8M urea buffer and then digested together with soluble proteins for the label-free proteomics quantification. This approach will not give you the full information you might have gotten if you separate them. The unknown distribution of individual proteins between the sequestered (w/o function) pool and soluble (w/ function) pool might complicate functional annotation. Would the authors consider including the lists of significantly regulated genes and proteins identified by RNA-seq and proteomics analyses in supplemental data? These would provide essential details of volcano plots and heatmap.

Answer: We thank for this helpful comment. We agree that a two-step protocol separating soluble and insoluble protein pools would have been more informative. Having said this, we were concerned that incomplete homogenization of the heart tissue would “trap” variable amounts of soluble protein in the insoluble fraction and increase variability between the samples. As the primary aim of the proteome study was to correlate protein and transcript data, we had opted for the simpler and more complete extraction protocol.

We further agree that the lists underlying the condensed figures of the omics datasets provide essential information. We have added the cardiac proteome data from 2- and 5-week-old mice as Supplementary tables 6 and 7, respectively. We have also included the lists of significantly differentially regulated genes from 2- and 5-week-old mice in the revised Supplementary information (Supplementary Data 5 and 6).

2) In Fig 6A, insoluble pellet fractions were isolated by differential centrifugation of cardiac protein extracts from both CAG-hBAG3^{P209L} mice and CAG-hBAG3^{WT} mice. The protein aggregates were implied to be the predominant component of these insoluble fractions. Were the pellets collected in largely reduced yields from the CAG-hBAG3^{WT} mouse hearts when protein aggregates were not observed in hearts of the CAG-hBAG3^{WT} mice at 5 wks of age (Suppl. Fig. 10C-F)?

Answer: The pellets of the differential centrifugation mainly consist of insoluble myofibrillar proteins. As these are the main component of cardiomyocytes we did not

observe any detectable differences between the pellets of hearts containing aggregates and hearts without aggregates.

In Fig, 6B. It is not clear how the “relative amounts” of hBAG3 protein and endogenous mBAG3 protein were defined. What was used to normalize the WB band intensities as relative ratios of protein expression?

Answer: We have used Coomassie-stained gels for normalization because myosin heavy chain isoforms, which were our loading control, are strongly induced in CAG-BAG3^{P209L} hearts and could therefore not be used for normalization. We have added these blots to the Source data.

A detailed protocol of differential fractionation (or citation) is missing. Some quality control validation would be helpful to demonstrate an efficient separation of protein aggregates from other insoluble proteins (e.g., cytoskeleton, contractile proteins).

Answer: Quality controls were made by detection of myosin heavy chain isoforms for the pellet fraction and GAPDH for the soluble fraction. We have included the respective blots in the Source data. We also provide a more detailed protocol for differential fractionation in the revised Methods section (pages 34-35, lines 766-793).

3) Suppl. Fig. 9F, the P209L panel showed multiple IB bands at lower M.W. positions. Are they fragments of hBAG3^{P209L} protein? If so, these protein fragments exist in purified hBAG3^{P209L} samples. What could be the possible cause for generating these fragments? Were these low M.W. bands included in quantification?

Answer: Sedimented proteins were identified by Coomassie-stain. It is most likely that the bands at lower molecular mass represent proteolytic fragments of the recombinant proteins, as these often appear during purification. They were included in the quantification, as can be seen in the Source Data file.

4) On Page 24, L521-523: the Statement “Mice ... did not display any obvious phenotype despite the strong hBAG3^{WT}-eGFP expression” should be supported by Suppl. Fig. S10 instead of S9.

Answer: We thank the reviewer for picking up on this and have changed this accordingly.

5) Fig. 7B. The last lane of WB shows almost no mCherry expression; it is unclear whether shRNA was effective in this sample. It would be more informative if the loading controls were included.

Answer: We thank the reviewer for this comment and have performed additional shRNA-treatment experiments with our CAG-BAG3^{P209L}-mice. We have performed additional experiments and present more control and treated samples in Fig. 7B and have added beta-actin loading controls. We now show immunoblot data from 6

scrambled shRNA and 6 BAG3-shRNA treated mice including loading controls and their quantification. In these new samples the AAV transfection rates proved higher, as can be seen by the relatively uniform mCherry expression. Overall, our collected data reflect the biological and technical variability, which is to be expected after such treatment.

6) The manuscript could use another round of editing to improve clarity. Multiple typos, inconsistent abbreviations, and mislabeling were noticed. Several examples are listed below, Both “ α MHC-BAG3^{WT}-eGFP” and “ α MHC-BAG3^{WT}” were used representing the same mouse line; the same issue for “ α MHC-BAG3^{P209L}-eGFP” and “ α MHC-BAG3^{P209L}”.

Answer: We have gone carefully over the ms and re-edited it, where appropriate. We have changed the name of the mouse lines accordingly and used this consistently throughout the manuscript and figures: α MHC-mouse lines were designated as α MHC-BAG3^{WT} and α MHC-BAG3^{P209L}.

Fig. 1B-D legend “Langendorff-isolated CMs from 10 weeks old WT, α MHC-BAG3^{WT}, and α MHC-BAG3^{P209L} mice stained for α -actinin (white) displaying Z-disc localization of BAG3^{WT} (insert)”: Please consider separating legends for (B) and for (C-D); WT strain was used in (C-D) only; does “insert” mean “zoomed region”?

Answer: The legends were changed and split accordingly, and “insets” defined as zoomed regions”.

Fig. 1B legend: Details describing the figure would be helpful, “While BAG3^{WT} (green, top panel) formed a grid-like structure (zoomed region, top panel), BAG3^{P209L} (green, bottom panel) was found to form aggregates of different sizes (arrows, bottom panel).”

Answer: The legend has been changed, as suggested by the reviewer.

Fig. 1F. “Immunogold stainings (black dots)” were conducted only with α MHC-BAG3^{P209L} mouse heart. This image shows that sarcomeric disarray was observed specifically in eGFP-positive but not in eGFP-negative CMs of the same heart.

Answer: The legend has been revised accordingly; “from WT” was deleted.

According to main text (P6, Lines 105-108), the eGFP+ CMs were isolated from both α MHC-BAG3^{WT} and α MHC-BAG3^{P209L} Tg mouse hearts, respectively. Immunoblotting was conducted using the isolated CMs (Suppl. Fig. 1C-D). The current legend of Suppl. Fig. 1C-D mislabelled these samples as being obtained from the whole heart.

Answer: For quantification of the ratio of hBAG3-eGFP vs. mBag3, we isolated CMs with Langendorff perfusion and sorted for eGFP-positive CMs by flow cytometry. Sorted CMs were lysed and protein extracted. The legend has been changed accordingly.

Additional Points for Your Considerations:

Line 68: Remove comma after disruption.

Removed.

Line 69: Pick a simpler phrase for "in congruence with".

Replaced with "in agreement".

Italicize "in vitro"

Was changed accordingly.

Line 82: Is there a simpler word than "recapitulate", here and elsewhere? Perhaps "mimic" or "imitate"?

Replaced by "mimics" here and in the abstract.

Line 94: Should there be a hyphen in "10 weeks" if there is one after "3"?

Hyphen added.

Line 96: It could benefit readability to be consistent with using an Oxford comma or not (putting a comma before "and" in a list). Elsewhere, it was used, but here it wasn't.

Comma was added.

Line 127: "hBAG3 expression" would read better than "expression of hBAG3". This rule can be applied elsewhere when deemed helpful.

Was changed accordingly. Lines 124, 128, 391, 425, 452.

Line 136: Add a hyphen between "eGFP" and "marked" if the structures are marked with eGFP.

Hyphen added.

Line 137: Add a comma after "antibody" before "recognizing" or clarify what the second part (with "recognizing") is referring to.

Comma added.

Line 149: Do "neither presented" instead of "did neither present".

Changed accordingly.

Line 161: "Bred to homozygosity with PGK-Cre" may read better.

Changed accordingly.

Line 163: Do "mild" and "variable" contradict each other? Or is there a wide enough range for "mild" to still be variable?

We agree with the reviewer and have deleted "overall mild".

Line 168: Add a comma before "displaying". As is, it could read that the

littermates displayed growth retardation, not that the CAG-BAG3 mice displayed growth retardation.

Comma added.

Line 170: "euthanized" in lieu of "sacrificed".

Replaced.

Line 172: Choose a consistent hyphenation pattern.

The hyphens at body weight and heart weight have been removed.

Line 177: "on hearts" reads better than "of hearts". Or it would be better to put "hearts" at the end of the sentence ("fluorescence on 1 to 5 week old CAG-BAG3P209L mices' hearts.").

Changed accordingly.

Line 180: If scientifically accurate, "Transgene expression in hearts" would be better and more readable than "The expression of the transgene in hearts".

Changed accordingly.

Line 189: Add "that" after "found".

Changed to "that the number of enlarged CMs (Fig. 2E, arrows) increased with age".

Line 198: "caused by collagen deposits replacing apoptotic cells" reads easier. It removes a "by" so it is more straightforward.

Changed accordingly.

Line 258: Change to "indicative of".

Changed accordingly.

Reviewer #3 (Remarks to the Author):

I have carried out a technical review of the RNAseq in this manuscript.

The authors generated a transgenic mouse line with conditional expression of hBAG3P209L-eGFP under the control of the CAG-promoter (CAG-BAG3P209L), which was found to be exclusively expressed in CMs by immunofluorescence staining.

They performed single-cell RNA-seq on the control and CAG-BAG3P209L mice at 2 weeks of age, to determine the ratio of hBAG3 to mouse Bag3 expression in eGFP+ CMs. Subsequently, they also performed bulk RNA-seq on 50-70 CMs, and tissue RNA-seq to compare CAG-BAG3P209L to control mice.

1. It is not mentioned how many cells were included in single-cell RNA-seq.

Answer: For the single-cell RNA-seq analysis the following cell numbers were included: mouse #26 WT: 24 cells; mouse #73 WT: 9 cells; mouse #25 CAG-BAG3^{P209L}: 21 bright cells and 18 dim cells; mouse #116 CAG-BAG3^{P209L}: 59 bright cells and 50 dim cells. Total = 181 cells. The "bulk RNA-seq" on 50-70 CMs was performed as a cluster analysis from the single-cell RNA-seq data. Clusters of single cells from CAG-BAG3^{P209L} mice (n= 80 bright CMs) and control mice (n= 33 CMs)

were pooled and analyzed for differentially expressed genes by using FindMarker function in Seurat V3.1. Since the term “bulk RNA-seq” for this type of analysis is misleading, we have changed it to “...differential expression analysis for our scRNA-seq from clusters...” (page 13, lines 264-265). The analysis approach is now also described in more detail in the revised Methods section (page 31-32, lines 697-704).

2. In methods (page 30), it is mentioned that the sequencing reads were mapped to the mouse genome. It is unclear how the ratio of human BAG3 to mouse Bag3 was determined if the reads were only mapped to the mouse genome. It is also unclear how the ratio was determined (was it based on the total number of reads mapped? Was it normalized by the gene length, etc?) since methods only state that the expression ratio was plotted using VlnPlot feature.

Answer: We have added more details to the revised Methods part “single-cell RNA-Seq analysis”. Human BAG3 and eGFP coding sequences were added to the mouse genome to serve as the template for mapping reads.

The gene expression counts were normalized using the default parameters in the Seurat package. Specifically, gene expression counts for each cell were divided by the total counts for that cell and multiplied by the scale factor, which was then natural-log transformed using log1p.

3. On page 12, “These experiments showed that the ratio of hBAG3 to mouse Bag3 was in the range of 0.5 to 2-fold, ... (Suppl. Fig. 7A)”. In Supplementary Figure 7A, the range of y-axis is set from log2 value of 1, which is equivalent to 2 fold ratio. This would mean hBAG3 to mouse Bag3 ratio was more than 2 fold in all cells, even in the control mice without the transgene. This also does not correspond to the range (0.5-2 fold) quoted in the main text.

Answer: We agree with the reviewer’s criticism and have re-plotted the ratio of hBAG3 to mouse Bag3 without a log2 transformation (new Suppl. Fig. 7A). To improve the clarity of the figure, we have inserted two breaks in the y-axis. The data now match the range mentioned in the text on page 12.

Moreover, it is unclear why some cells showed a high ratio of hBAG3/mBag3 in control mice.

Answer: scRNA-seq data typically have a higher level of technical noise such as dropout events and high gene expression background than bulk RNA-seq. The identification of very few eGFP and hBAG3 reads in control-samples could be due to background issues and the bioinformatic analysis process such as the mapping of reads.

4. List of differentially expressed genes should be reported as a supplementary data.

Answer: A list of significantly differentially expressed genes is now included in the Supplementary information as Supplementary Data 1 and 3.

5. Significance of GO-analysis should be reported as a supplementary data (size of gene sets, level of significance, list of genes included, etc.)

Answer: Lists of KEGG and GO-terms for bulk RNA-seq have been added to the Supplement as Supplementary Data 2 and 4.

6. Authors reported 35 downregulated genes from bulk RNA-seq of a pool of 50-70 individual CMs, while 1854 downregulated genes were reported from the tissue RNA-seq. From how many mice were the 50-70 CMs obtained?

Answer: As mentioned above, this was a cluster analysis from the single-cell RNA-seq data and therefore Bonferroni corrected. The dataset which was used for this type of analysis contained 2 CAG-BAG3^{P209L} and 2 wt mice (#26 WT, #73 WT, and #25 CAG-BAG3^{P209L}, #116 CAG-BAG3^{P209L}), we have added this information to the revised Methods section (page 32, lines 701-704).

How many mice were included in tissue RNA-seq?

The difference between the number of downregulated genes is striking and maybe many of the downregulated genes detected in tissue RNA-seq arises from the differences between mice. Reporting the number of replicates included in tissue RNA-seq would minimize confusion around this.

Answer: We thank the reviewer for making this important point. We analyzed 6 control and 6 CAG-BAG3^{P209L} mice. This information has been added to the respective legend in Fig.5 and to the revised Methods section (page 30, lines 670-671).

7. Minor spelling errors (e.g. “Volcano blot” in page 45)

Answer: Was corrected to “Volcano plot”.

8. The color scale of Figure 5B indicates “log2(fold of mean)”. Are the values in the heatmap fold changes, or actual expression values?

Answer: These are transformed expression values from a DeSeq2 normalized count table and show the Z-score. The legend in Fig. 5B was changed accordingly from “log2 (fold of mean)” to “Expression Z-score”.

Out of 3804 differentially expressed genes, how were the genes shown in Figure 5B selected? From It should also be discussed clearly

Answer: The genes were selected from the output of the GO-analysis and manually for the “Fetal gene program” cluster. It consists of the following genes: *Myh6, Atp2a2, Atp2b4, Hand2, Gata6, Mef2c, Nkx2-5, Gata4, Acta1, Lmna, Myh7, Nppa, Nppb*. This information has been added to the legend of Fig. 5B.

Reviewer #4 (Remarks to the Author):

Kimura et al – referee report Nat Comm

In their manuscript, Kimura et al report on the phenotypes of a humanized transgenic BAG3P209L mouse models that they generated. The mice show massive fibrosis resulting in a severe, early-onset restrictive cardiomyopathy with increased mortality and histological analyses shows Z-disc disintegration and formation of protein aggregates, all similar to what has been observed in patients.

Previous attempts by others had failed to reveal such a phenotype, which -as the authors suggest- may have been due to too low expression levels. In fact, the authors confirm this idea and now show that mice with lower expression do already show protein aggregation and mild morphological changes, but do not (yet) present signs of cardiomyopathy, compromised heart function and early mortality. A second model, however, in which higher expression levels were achieved, shows not only more (rapid) aggregation and more morphological damage to sarcomeric structures, but also a clear functional phenotype. Besides early onset skeletal muscle weakness, these mice displayed clear signs of cardiomyopathy, and at 5 weeks of age suffered from severe heart failure or died prematurely. This is totally in line with the progressive character of this disease in humans. It also supports the concept that progressive protein aggregation drives this disease. The data on this part are clear, novel, and valuable as these mice may hence be a important tool for evaluating therapeutic strategies for this disease.

Even more so, as BAG3 levels are upregulated by different forms of stress, the current data also have relevance to preventive measures in these patients (i.e. to avoid physical stress), which is a message that could be more strongly conveyed.

Using RNA-Seq analyses in 5-week-old mice, the authors next show a number of differentially expressed genes, consistent with the degenerative features in the cardiomyocytes (lower expression of genes involved in energy metabolism, oxidative phosphorylation and cardiac muscle contraction) and the presence of a proteotoxic stress (upregulation of genes various (protein) quality control pathways). Proteomic analysis nicely corroborated these findings. Whilst interesting as observation and the data are well presented, these data add little information in terms of cause-consequence relations, except for the fact that many of the changes found involve BAG3-interactors. The latter indeed would be consistent with (but by themselves provide no proof for) suggestions from previously published data on a BAG3-driven protein aggregation process that has triggered a rather broad protein stress response (Hsp's, autophagy), maybe as an attempt to restore protein homeostasis.

More revealing in terms of mechanisms are their data to show that the mutant human BAG3P209L associated normally with its partners HSPA8 and HSPB8 (i.e similar to BAG3WT) but accumulates in insoluble protein aggregates (as revealed by both biochemical and immunofluorescent, FRAP analyses) in which also the endogenous, wildtype mouse BAG3 is sequestered. As correctly referenced by the authors, this is completely consistent with previously published work in cell models. So, whereas these data do not bring in a novel concept, they are important as an in vivo validation of the published cell biological work and supports the idea that also at the organismal levels the

BAG3P209-related disease progression can be linked to a progressive collapse of protein homeostasis.

Finally, the authors show how gene therapy using AAV-mediated knockdown of hBAG3 strongly mitigates the disease phenotype. Whereas well-executed and showing the proof-of-concept that AAV strategies can lead to a knockdown of a transgenic human BAG3P209L in mice and reverse its toxicity (which is not really a surprise), I find their conclusion that this demonstrates ‘that an AAV-based gene therapy is a feasible and promising therapeutic strategy to halt or even reverse the progression of this severe myofibrillar myopathy’ (discussion) premature. The crux of the problem for such a strategy lies in the none-specificity of such an approach that also would target the wildtype allele. This likely will result in major toxicity issues, given the importance of BAG3 for cellular physiology. I am aware of and do appreciate the fact that the authors themselves also write that shRNA specific to the mRNA carrying the point mutation has to be identified. However, this seems easier said than done and as is true for the suggested alternative of CRISPR-based gene-editing tools. So, here their ‘promising’ statement should be tempered.

Answer: We thank the reviewer for his/her appreciation of our work and the constructive criticisms to further improve presentation and writing.

We agree with the relevance of stress and have implemented this point into the revised version of the manuscript by adding this sentence to the discussion section (page 22, lines 463-465): “...and to protect from mechanical stress. As a consequence, patients suffering from MFM6 should avoid physical exertion such as strenuous exercise. This could lead to progressive muscle damage, as has been observed in other MFM models²².”

We also concur with the reviewer in regard to technical issues and limitations of identifying a shRNA specific against a point mutation (Schwarz et al., 2006). We also agree with the reviewer that knockdown of endogenous Bag3 would most likely lead to major toxicity issues, as we also mentioned in the Discussion section. We have therefore rephrased this sentence and emphasize that it could be “...theoretically possible to halt or even reverse...”. We now explain that either micro-RNAs (Acunzo et al., 2017) or a CRISPR/Cas9-based approach such as prime editing could be tested as alternatives. We have added the potential limitations and drawbacks of these approaches in the revised version of the Discussion section (page 22, lines 476-481).

Schwarz DS, Ding H, Kennington L, Moore JT, Schelter J, Burchard J, et al. Designing siRNA that distinguish between genes that differ by a single nucleotide. *PLoS Genet.* 2006;2(9):e140.

Acunzo M, Romano G, Nigita G, Veneziano D, Fattore L, Laganà A, et al. Selective targeting of point-mutated KRAS through artificial microRNAs. *Proc Natl Acad Sci U S A.* 2017;114(21):E4203-e12.

Minor comments

Introduction:

- **To my knowledge BAG3 does NOT balance protein synthesis in response to stress but it does alters/increases protein degradation in response to stress, but also not exclusively upon mechanical stress (although this could be a major physiologically relevant stress). This should be revised.**

Answer: We thank the reviewer for this thoughtful comment. Published work illustrates that BAG3 interacts with components of the mTORC1 signalling pathway and in this way regulates mTORC1 activity (Kathage et al., 2017). In turn, mTORC1 affects protein translation and autophagy. It was further demonstrated that BAG3 overexpression/depletion affects the phosphorylation of the mTORC1 target and ribosomal subunit S6 and thereby modulates efficiency of protein translation (Kathage et al., 2017). Association of BAG3 with the mTORC1 inhibitor TSC1 is regulated by mechanical stress (Kathage et al., 2017). We therefore feel that it is correct to describe BAG3 as a proteostasis factor that balances protein synthesis and protein degradation in response to mechanical stress.

- **There is no evidence to show that BAG3 itself recognizes mechanically unfolded and damaged forms of FLNC. Rather, data suggest that in conjunction Hsp70 proteins, it targets Hsp70 recognized client proteins to autophagy and does so under conditions of proteasomal overload, conditions by which it is upregulated. This is also why CASA as terminology for this process is confusing. Moreover, albeit that HSPB8 is a preferred small HSP partner of BAG3, HSPB8 is dispensable for the BAG3dependent autophagosomal targeting of clients (see the work of Carra and co-workers on this matter).**

Answer: We thank the reviewer for pointing out this inaccuracy in the wording. We agree that recognition of mechanically unfolded and damaged forms of FLNC is mediated by BAG3 in conjunction with its partner chaperone Hsp70. The role of HSPB8 is still a matter of debate: While Carra and colleagues presented data that autophagosomal targeting of clients can occur independently of HSPB8 (e.g. Carra et al., 2008), more recent data, nevertheless, demonstrate a close interplay of HSPB8 and BAG3 in the sequestration of ubiquitinated client proteins (Guilbert et al., 2018). Moreover, BAG3 interacts with small heat shock proteins (sHSPs) other than HSPB8, e.g. HSPB1 and HSPB5 (Rauch et al., 2017). Accordingly, these sHSPs might cooperate with BAG3 in the absence of HSPB8. Thus, we feel that the term chaperone-assisted selective autophagy (CASA) correctly reflects the close interplay of BAG3 with HSP70 and sHSPs proteins during client recognition and targeting for degradation.

1. Carra S, Seguin SJ, Lambert H, and Landry J. HspB8 chaperone activity toward poly(Q)-containing proteins depends on its association with Bag3, a stimulator of macroautophagy. *J Biol Chem.* 2008;283(3):1437-44.
2. Guilbert SM, Lambert H, Rodrigue MA, Fuchs M, Landry J, and Lavoie JN. HSPB8 and BAG3 cooperate to promote spatial sequestration of ubiquitinated proteins and coordinate the cellular adaptive response to proteasome insufficiency. *Faseb j.* 2018;32(7):3518-35.

3. Rauch JN, Tse E, Freilich R, Mok SA, Makley LN, Southworth DR, et al. BAG3 Is a Modular, Scaffolding Protein that physically Links Heat Shock Protein 70 (Hsp70) to the Small Heat Shock Proteins. J Mol Biol. 2017;429(1):128-41.

Results

• ...the sarcomeric protein titin, which is not under control of BAG3-mediated proteostasis....

Proteostasis refers to protein homeostasis and not protein quality control only.

Please rephrase to : ...the sarcomeric protein titin, which is not under control of BAG3-dependent protein quality control.....

Answer: We have changed the sentence accordingly.

• **It is very interesting though that titin is not affected, whilst FLNC and desmin are:**

To me this does not imply that “diverse Z-disc components are subjected to distinct aggregation pathways”, but rather that they may depend on different protein quality control systems for their integrity or that titin is more stable/less affected by the aggregation cascade initiated by BAG3^{P209L}. Besides my request to rephrase this sentence accordingly, I wonder whether the authors could speculate why this could be different.

Answer: We thank the reviewer for this comment. We agree that different quality control systems might exist for diverse Z-disc components. Yet, titin spans the whole sarcomere and hence cannot be considered solely as a Z-disc protein. We agree that titin is more stable and less dynamic than FLNC and desmin and less affected by the BAG3^{P209L} mutation. The text has been revised accordingly.

I wonder whether the authors could speculate why this could be different

Answer: Titin spans the whole sarcomere and only subdomains are located at the Z-disc. Force-induced unfolding has been described for titin domains that are localized in I-bands adjacent to Z-discs and quality control of these domains involves the Hsp90 chaperone system. This might explain why titin is more stable and less affected by a functional impairment of the Z-disc-associated BAG3 protein quality control system compared to FLNC and desmin.

REVIEWERS' COMMENTS

Reviewer #1 (Remarks to the Author):

The reviewers have effectively responded to all the reviewer comments and the manuscript is vastly improved based on the modifications.

Reviewer #2 (Remarks to the Author):

The authors made a thorough round of editing during their revision, which largely improved the clarity and coherence of the manuscript. The revised conclusions are better supported by your current data. I understand your concern of changing the sample collection protocol. Hope such technical challenges could be overcome in future studies. A few minor issues remain and require your consideration.

1) The claim “the disease mechanism is due to...” of the abstract (P3, Line 44) might be a little bit strong.

2) (P12, Line 242) Is Table 1 also in the supplementary Information?

3) For supplementary information, a table of content will be helpful for readers to locate figures and tables. In particular for tables related to updated supplementary data.

4) For tissue imaging data, the scale bar sometimes blends into the background (e.g., Fig 7d&7f). Maybe consider using bright color to improve contrast. Should the size of scale bars also be included?

Reviewer #3 (Remarks to the Author):

My comments especially focussed on the bioinformatics dataset have been adequately addressed.

Reviewer #4 (Remarks to the Author):

2021_03_08 – referee report revision Kimura et al – referee report Nat Comm, Reviewer #4

As stated before, this paper reports on a nice mouse model that recapitulates what is seen in humans carrying a Bag3P209L mutations; hence this model could be an important tool for evaluating therapeutic strategies for this disease.

In response to my initial comments, the authors now have included the putative relevance of BAG3 being upregulated by different forms of stress and preventive measures to be taken in these patients, which I think truly is an important message to convey. They have also now adequately tempered their statements related to the use of CRIPSR-based gene-editing as promising therapeutic option.

Regarding to my specific questions:

The last 3 were adequately dealt with, but I think that the first two still require attention

1) I disagree with their answer of the role of BAG3 in protein synthesis. Here, a distinction should be made between direct involvement (as they now suggest) and indirect consequences. This is often a problem related to the cell biological effects of chaperone modulation or defects. Whereas a clear direct involvement of the BAG3 -related chaperone complex in stress-induced degradation (autophagy) has been demonstrated, its direct involvement in translational inhibition has not (even though occurring as a downstream event). Their text on this must be changed as it may incorrectly suggest that it could be related to the stress-granule biology that also occurs upon the identical stresses that activate BAG3, but that seems more to engage and depend on its partner HSPB8 rather than on the action of BAG3 itself.

2) Despite that the authors agree with my comment on the inaccuracy in the wording that relate to BAG3 recognizing mechanically misfolded proteins, they have not adapted their text. Moreover, whereas I agree that BAG3 is (indirectly) involved in the sequestration of ubiquitinated client proteins (indeed re-routing them from the proteasomal to autophagosomal degradation) , I also still disagree with their reply that, because BAG3 also can interact with other sHSPs, that these could be involved in the PROCESS of autophagy (i.e. chaperone-assisted selective autophagy): to my knowledge there is no direct experimental evidence showing this to be the case.

Point-by-point response to Reviewer Comments

Reviewer #1 (Remarks to the Author):

The reviewers have effectively responded to all the reviewer comments and the manuscript is vastly improved based on the modifications.

Reviewer #2 (Remarks to the Author):

The authors made a thorough round of editing during their revision, which largely improved the clarity and coherence of the manuscript. The revised conclusions are better supported by your current data. I understand your concern of changing the sample collection protocol. Hope such technical challenges could be overcome in future studies. A few minor issues remain and require your consideration.

1) The claim “the disease mechanism is due to...” of the abstract (P3, Line 44) might be a little bit strong.

Answer: The claim was weakened to: “Our data suggest that the disease mechanism is due...”

2) (P12, Line 242) Is Table 1 also in the supplementary Information?

Answer: We are sorry for the confusion but there is only one supplementary table. We changed “Table 1” to “Supplementary Table 1” (p12, line 226 in revised manuscript).

3) For supplementary information, a table of content will be helpful for readers to locate figures and tables. In particular for tables related to updated supplementary data.

Answer: We will add a table of contents after consultation with the editors if it suits the style of the journal.

4) For tissue imaging data, the scale bar sometimes blends into the background (e.g., Fig 7d&7f). Maybe consider using bright color to improve contrast. Should the size of scale bars also be included?

Answer: Scale bars in Fig. 7c-f have been changed to 1.5 pt. Scale bar in 7f was changed to black. Sizes of scale bars is in the figure legends and, according to the journals style, are not in the figures.

Reviewer #3 (Remarks to the Author):

My comments especially focussed on the bioinformatics dataset have been adequately addressed.

Reviewer #4 (Remarks to the Author):

2021_03_08 – referee report revision Kimura et al – referee report Nat Comm, Reviewer #4

As stated before, this paper reports on a nice mouse model that recapitulates what is seen in humans carrying a Bag3P209L mutations; hence this model could be an important tool for evaluating therapeutic strategies for this disease.

In response to my initial comments, the authors now have included the putative relevance of BAG3 being upregulated by different forms of stress and preventive measures to be taken in these patients, which I think truly is an important message to convey. They have also now adequately tempered their statements related to the use of CRIPSR-based gene-editing as promising therapeutic option.

Regarding to my specific questions:

The last 3 were adequately dealt with, but I think that the first two still require attention

1) I disagree with their answer of the role of BAG3 in protein synthesis. Here, a distinction should be made between direct involvement (as they now suggest) and indirect consequences. This is often a problem related to the cell biological effects of chaperone modulation or defects. Whereas a clear direct involvement of the BAG3 - related chaperone complex in stress-induced degradation (autophagy) has been demonstrated, its direct involvement in translational inhibition has not (even though occurring as a downstream event). Their text on this must be changed as it may incorrectly suggest that it could be related to the stress-granule biology that also occurs upon the identical stresses that activate BAG3, but that seems more to engage and depend on its partner HSPB8 rather than on the action of BAG3 itself.

Answer: The text in the introduction was changed accordingly. We have deleted the statement that BAG3 balances protein synthesis in response to mechanical stress and now focus on the role of BAG3 in mediating protein degradation in response to stress, such as mechanical strain.

2) Despite that the authors agree with my comment on the inaccuracy in the wording that relate to BAG3 recognizing mechanically misfolded proteins, they have not adapted their text. Moreover, whereas I agree that BAG3 is (indirectly) involved in the sequestration of ubiquitinated client proteins (indeed re-routing them from the proteasomal to autophagosomal degradation) , I also still disagree with their reply that, because BAG3 also can interact with other sHSPs, that these could be involved in the PROCESS of autophagy (i.e. chaperone-assisted selective autophagy): to my knowledge there is no direct experimental evidence showing this to be the case.

Answer: We have amended the text in the Introduction as requested.